# Non-canonical pathway for Rb inactivation and external signaling coordinate cell-cycle entry without CDK4/6 activity

Mimi Zhang [1,3], Sungsoo Kim [1,2,3] & Hee Won Yang [1,2] ✉

Cyclin-dependent kinases 4 and 6 (CDK4/6) are critical for initiating cell proliferation by inactivating the retinoblastoma (Rb) protein. However, mammalian cells can bypass CDK4/6 for Rb inactivation. Here we show a non-canonical pathway for Rb inactivation and its interplay with external signals. We find that the non-phosphorylated Rb protein in quiescent cells is intrinsically unstable, offering an alternative mechanism for initiating E2F activity. Nevertheless, this pathway incompletely induces Rb-protein loss, resulting in minimal E2F activity. To trigger cell proliferation, upregulation of mitogenic signaling is required for stabilizing c-Myc, thereby augmenting E2F activity. Concurrently, stress signaling promotes Cip/Kip levels, competitively regulating cell proliferation with mitogenic signaling. In cancer, driver mutations elevate c-Myc levels, facilitating adaptation to CDK4/6 inhibitors. Differentiated cells, despite Rb-protein loss, maintain quiescence through the modulation of c-Myc and Cip/Kip levels. Our findings provide mechanistic insights into an alternative model of cell-cycle entry and the maintenance of quiescence.

The precise regulation of cell-cycle entry represents a fundamental and vital process within multicellular organisms. Dysregulation of this process has been implicated in various pathological conditions, including cancer, developmental disorders, and degenerative diseases[1,2]. Cyclin-dependent kinases 4 and 6 (CDK4/6) and the retinoblastoma (Rb) protein play key roles in the initiation of the cell cycle[3]. During quiescence, non-phosphorylated Rb directly binds and inhibits E2F transcription factors, which are crucial regulators of cell-cycle genes, thereby restraining cell-cycle entry[4]. In the canonical model of cell-cycle entry, mitogenic signaling increases cyclin D expression to activate CDK4/6[5] (Fig. 1a left). In turn, active CDK4/6 phosphorylates and inactivates Rb, resulting in E2F activation for cell-cycle entry[6–10]. In the context of cancer, alterations in CDK4/6 activity through gene mutations, deletions, and amplifications are frequently observed[1,2]. CDK4/6 inhibitors (CDK4/6i) have emerged as first-line therapeutic strategies for metastatic hormone receptor-positive (HR⁺) breast cancer and demonstrate promising potential in other malignancies such as non-small-cell lung cancer and melanoma[11–13].

Suppression of CDK4/6 activity, triggered by conditions such as limited mitogens, contact inhibition, CDK4/6i treatment, activation of the p53/p21 pathway, or increased p16 expression due to aging, typically halts cell proliferation. However, previous studies show that non-transformed mammalian cells can bypass the requirement for CDK4/6 activity to progress through the cell cycle[14–17]. This has been shown by cell proliferation in mice lacking all cyclin D1/2/3 isoforms or CDK4/6[14,15] and the initiation of DNA replication without CDK4/6 activity and Rb phosphorylation[16]. Furthermore, another study revealed that cell growth can induce the dilution of Rb concentration, even without a decrease in the overall Rb levels[17]. The transcription factor c-Myc has been posited as a potential driver of this non-canonical pathway by directly promoting the expression of the CDK2 activator cyclin E[16,18]. Nonetheless, the cyclin E promoter lacks consensus c-Myc binding sites[19], and the mechanism by which cell growth leads to the specific dilution of Rb protein remains elusive.

In this study, we elucidate a non-canonical pathway for Rb inactivation and delineate how other cell-cycle regulators coordinate cell

[1]Department of Pathology and Cell Biology, Columbia University, New York, NY 10032, USA. [2]Herbert Irving Comprehensive Cancer Center, Columbia University, New York, NY 10032, USA. [3]These authors contributed equally: Mimi Zhang, Sungsoo Kim. ✉e-mail: hy2602@cumc.columbia.edu

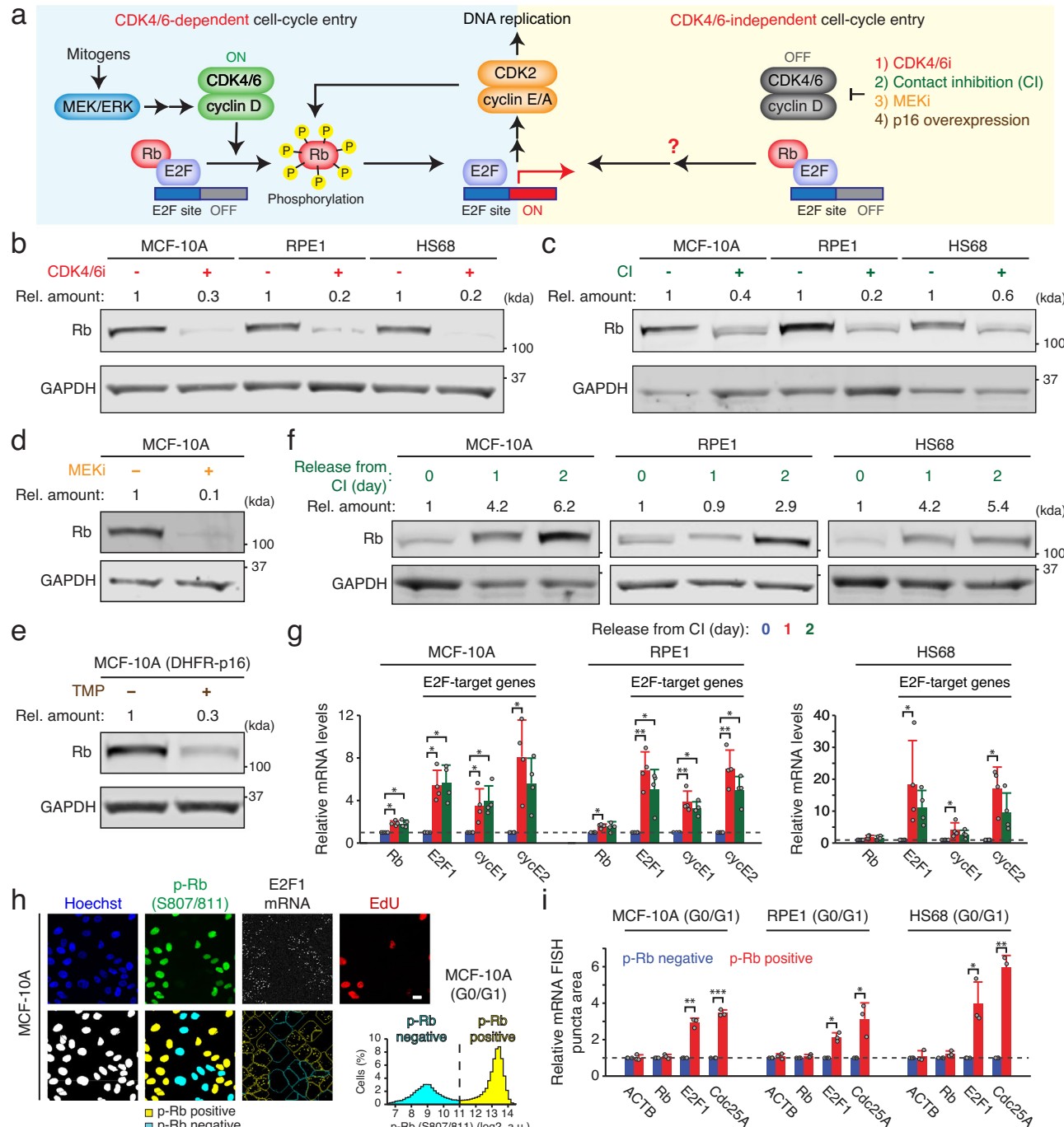

**Fig. 1 | Reduced Rb-protein levels after CDK4/6 inhibition. a** Schematic diagram depicting the two distinct pathways involved in cell-cycle entry. **b**–**e** Immunoblot showing Rb and loading control GAPDH expression levels 48 h after various CDK4/6 inhibition methods: palbociclib (1 μM) (**b**), contact inhibition (**c**), trametinib (10 nM) (**d**), or p16 overexpression induced by TMP (50 μM) (**e**). Relative amounts indicate Rb-band intensities normalized with GAPDH ($n \geq 2$ biological replicates). **f** Immunoblot showing Rb and GAPDH expression in cells released from 48 h-contact inhibition ($n \geq 2$ biological replicates). **g** Relative mRNA levels of Rb and E2F-target genes in cells released from 48 h-contact inhibition. Data are shown as mean ± SD ($n = 4$ biological replicates). Asterisks indicate significant differences in the one-way ANOVA test (*$p \leq 0.05$; **$p \leq 0.001$). **h** Representative images of Hoechst, p-Rb at S807/811, E2F1 mRNA FISH, and EdU staining in MCF-10A cells. The scale bar is 20 μm. A histogram of p-Rb in G0/G1-phase MCF-10A cells shows classification of p-Rb positive and negative cells. **i** Relative mRNA FISH levels of ACTB, Rb, E2F1, and CDC25A in G0/G1. Data are shown as mean ± SD ($n = 3$ biological replicates). Asterisks indicate significant differences in the two-tailed unpaired t-test (*$p \leq 0.05$; **$p \leq 0.001$; ***$p \leq 0.0001$).

proliferation in the absence of CDK4/6 activity. Our findings unveil an alternative pathway for cell-cycle entry that incorporates both intrinsic and extrinsic signaling mechanisms. This alternative regulatory model serves to optimize cellular growth and maintain cellular integrity under conditions where CDK4/6 activity is inhibited, shedding light on the complex interplay of factors that govern the cell cycle.

## Results

### CDK4/6 inhibition reduces expression levels of Rb protein

We investigated the impact of CDK4/6 inhibition on Rb expression in non-transformed mammary epithelial (MCF-10A) and retinal pigment epithelial (RPE1) cells, as well as passage-limited fibroblast (HS68) cells. We employed four distinct approaches to suppress CDK4/6

activity: the CDK4/6i palbociclib, contact inhibition, the MEK inhibitor trametinib (MEKi), and p16 overexpression (Fig. 1a right). Using the dihydrofolate reductase (DHFR)-trimethoprim (TMP) protein stabilization system[20], we induced p16 overexpression in MCF-10A cells that lack p16 expression[21]. This system leverages the binding of TMP to DHFR to rapidly stabilize the conjugated p16 protein, leading to a suppression of CDK4/6 activity within 3 h of TMP addition (Supplementary Fig. 1a, b). We found that CDK4/6 inhibition by all four different methods for 2 days resulted in a substantial reduction in Rb-protein levels (Fig. 1b–e). The reduction in Rb-protein levels (40–88%) was more pronounced than the decrease in Rb mRNA levels (4–44%) (Fig. 1b–e and Supplementary Fig. 1c–f). We confirmed that our Rb-protein measurements fall within a linear range (Supplementary Fig. 1g, h). To evaluate the reversibility of the Rb-protein reduction, we released cells from a 2-day contact inhibition or CDK4/6i treatment by either re-plating them at a lower confluency or washing out the drug. Rb-protein levels gradually rebounded as cells were liberated from contact inhibition or CDK4/6i treatment (Fig. 1f and Supplementary Fig. 1i). However, we observed a relatively modest upregulation of Rb mRNA levels (1.5–1.9-fold change) compared to the concurrent increase in mRNA expression levels of E2F-target genes (3–18-fold change) (Fig. 1g). These findings demonstrate that CDK4/6 inhibition leads to a substantial reduction in Rb-protein levels.

The *Rb* promoter contains a conserved binding site for the E2F transcription factors[22], which could explain the decrease in Rb expression following CDK4/6 inhibition. To examine whether E2F activity regulates Rb mRNA, we classified cycling MCF-10A cells based on Rb phosphorylation at Serine 807/811, a previously characterized marker for Rb hyperphosphorylation[10] (Fig. 1h). Antibody staining showed a bimodal distribution of nuclear intensity, where cells in the upper peak mostly exhibit Rb hyperphosphorylated and inactive. Given that atypical E2Fs, E2F7 and E2F8, can suppress E2F transcriptional activity during the S phase[23], we further selected G0/G1-phase cells based on Hoechst and 5-ethynyl-2´-deoxyuridine (EdU) staining. We performed mRNA fluorescence in situ hybridization (FISH) to measure mRNA expression levels of β-actin (ACTB, negative control), Rb, and two E2F-target genes, E2F1 and Cdc25A (positive control). While E2F1 and Cdc25A levels were significantly higher in cells with Rb phosphorylation at S807/811, both β-actin and Rb mRNA were expressed equally, irrespective of the Rb phosphorylation status in MCF-10A, RPE1, and HS68 cells (Fig. 1i). These results indicate that Rb mRNA is not directly regulated by E2F activity, and therefore the Rb-protein reduction induced by CDK4/6 inhibition is not attributable to the regulation of Rb mRNA.

## Non-phosphorylated Rb is unstable in quiescent cells

We tested whether the introduction of exogenous Rb, controlled by a constitutively active promoter, could prevent the Rb-protein reduction mediated by CDK4/6 inhibition. Using a lentiviral construct, we established MCF-10A and RPE1 cells stably expressing yellow fluorescent protein (YFP)-Rb. While exogenous YFP-Rb was expressed 1.7- and 5.3 times higher than endogenous Rb levels in MCF-10A and RPE1 cells, respectively, it did not affect CDK4/6-dependent cell-cycle entry (Supplementary Fig. 2a–d). In addition, despite high levels of YFP-Rb expression, CDK4/6i treatment near completely reduced both exogenous and endogenous Rb with similar kinetics within 24 h (Fig. 2a, b). CDK4/6 inhibition by two other methods, contact inhibition and mitogen removal, significantly reduced exogenous YFP-Rb levels compared to the control condition (Supplementary Fig. 2e, f). These results indicate that CDK4/6 inhibition equally reduces both endogenous and exogenous Rb-protein levels regardless of its expression levels and promoter activity.

We sought to determine whether CDK4/6 inhibition could reduce Rb-protein levels by enhancing global proteasome activity, as suggested by a previous study[24]. To test this, we simultaneously monitored the levels of nuclear localization signal (NLS)-mRuby3, histone 2B (H2B)-iRFP, and YFP-Rb. An increase in global proteasome activity should decrease the levels of NLS-mRuby3 and H2B-iRFP if it causes Rb-protein reduction. However, CDK4/6i treatment resulted in a decrease in YFP-Rb but neither NLS-mRuby3 nor H2B-iRFP levels (Supplementary Fig. 2g, h), indicating that the Rb-protein reduction is not due to increased proteasome activity.

To test whether Rb-protein levels are dependent on cell-cycle status, we established MCF-10A cells stably expressing YFP-Rb and a live-cell sensor for CDK2 activity[25]. After mitosis, daughter cells activate or inactivate CDK4/6 and subsequently CDK2 to enter the next cell cycle or quiescence, respectively[25–27]. We aligned single-cell traces by mitosis and classified them into proliferating and quiescent cells based on their CDK2 activity (Fig. 2c left). While continuously proliferating cells maintained high YFP-Rb levels, cells entering quiescence gradually reduced YFP-Rb levels (Fig. 2c middle and right). The half-life ($t_{1/2}$) of Rb protein in quiescent cells was 5 h (Fig. 2d). We proceeded to investigate whether reinforcing cell-cycle entry could prevent the Rb-protein reduction by CDK4/6 inhibition. Using the doxycycline-inducible system, we triggered cyclin E1 overexpression to drive CDK2 activation and cell proliferation in the presence of CDK4/6i. We found that CDK4/6 inhibition suppressed CDK2 activity and reduced YFP-Rb levels (Supplementary Fig. 2i, j). However, cyclin E1 overexpression induced CDK2 activation and prevented YFP-Rb reduction (Fig. 2e and Supplementary Fig. 2j). When we classified doxycycline-treated cells into proliferation and quiescence based on CDK2 activity, proliferating and quiescent cells induced relatively high and low YFP-Rb levels, respectively (Fig. 2e). To examine whether the stability of Rb protein is influenced by its phosphorylation status, we employed the protein synthesis inhibitor cycloheximide. To remove Rb phosphorylation, we treated cells with CDK4/6i for 24 h. The $t_{1/2}$ of Rb protein under the CDK4/6i-treated condition was approximately 4 times shorter than under the control condition (Fig. 2f). Furthermore, we found that the Rb phosphomimetic protein, where 14 CDK phosphorylation sites were replaced with glutamic acid, displayed about 5 times greater stability than the wild-type Rb in the presence of CDK4/6i (Fig. 2g). These results show the inherent instability of non-phosphorylated Rb in quiescent cells, which in turn leads to a decrease in its expression levels.

## Insufficient E2F activity by Rb-protein reduction alone

We examined whether the Rb-protein reduction alone could effectively induce E2F activation in the absence of CDK4/6 activity. To test this, we monitored the levels of Rb protein and E2F activity for 3 days following CDK4/6 inhibition in MCF-10A and RPE1 cells. We observed a substantial reduction in Rb-protein levels (~80%) within 1 day of CDK4/6 inhibition, which persisted throughout the duration of the experiment (Fig. 3a). However, despite this considerable reduction in Rb-protein levels, we found that these cells retained low E2F activity, as evidenced by the low mRNA expression levels of E2F target genes (Fig. 3b). Similarly, we found relatively low E2F activity compared to the control condition when CDK4/6 activity was inhibited through other strategies, contact inhibition or treatment with CDK4/6i or MEKi for 2 days (Supplementary Fig. 3a–c), despite noticeable reduction in Rb-protein levels (Fig. 1b–d). To monitor cell-cycle entry after CDK4/6 inhibition, we established MCF-10A cells expressing live-cell reporters for CDK4/6[26] and CDK2 activities as well as cell-cycle phases (Cdt1 degron)[28]. This system enables us to monitor distinct steps of cell-cycle entry. The Cdt1 degron detects G1/S and S/G2 transitions due to its sharp degradation during the S phase (Fig. 3c). While 67% of cells continuously activated CDK4/6 and CDK2 to initiate the cell cycle, CDK4/6i treatment suppressed CDK4/6 activity and subsequently CDK2 activity, halting cell proliferation within 24 h (Fig. 3d, e and Supplementary Fig. 3d). Single-cell data revealed that after a period of quiescence (G0), only 3% of MCF-10A

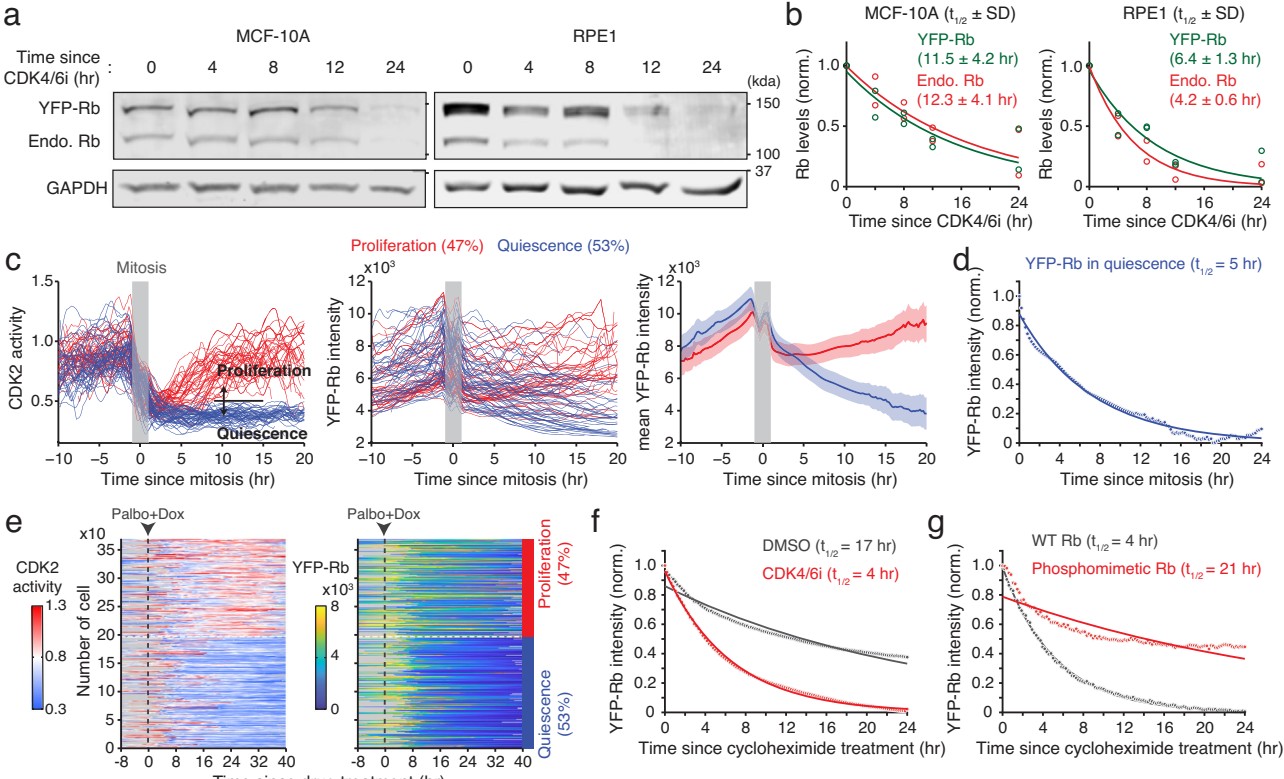

**Fig. 2 | Alterations in Rb-protein levels depending on cell proliferation.**
**a** Immunoblot showing exogenous and endogenous Rb and GAPDH expression in MCF-10A and RPE1 cells after treatment with palbociclib (1 μM). **b** Relative levels of endogenous and exogenous Rb protein. Solid lines indicate best-fitted lines. Rb half-life is shown as the mean ± SD (n = 2 biological replicates). **c** Single-cell traces of CDK2 activity and YFP-Rb levels and mean YFP-Rb levels aligned by the end of mitosis (anaphase) in MCF-10A cells. Based on CDK2 activity (threshold = 0.5, black line) between 9 and 15 h after mitosis, daughter cells were classified into proliferation or quiescence. Mean YFP-Rb levels are shown as mean ± 95% confidence interval (proliferation: n = 64 cells; quiescence: n = 72 cells). **d** Time course of relative YFP-Rb level change in MCF-10A cells entering quiescence. YFP-Rb levels

were normalized with the minimum level (n = 72 cells). **e** Heatmap of single-cell traces for CDK2 activity and YFP-Rb levels in MCF-10A cells treated with palbociclib (1 μM) + doxycycline (5 μM). Each row represents a single-cell trace over time according to the respective color map. During 25–40 h after drug treatment, cells activating CDK2 (>0.8) for over 2 h were classified into proliferating cells. **f** Time course of relative YFP-Rb level change in MCF-10A cells treated with cycloheximide (10 μg/ml). Cells were treated with either DMSO or palbociclib (1 μM) for 24 h before imaging (DMSO: n = 3613 cells; CDK4/6i: n = 2717 cells). **g** Time course of relative YFP tagged wild-type and phosphomimetic Rb in MCF-10A cells treated with cycloheximide (10 μg/ml). Cells were treated with palbociclib (1 μM) for 24 h before imaging (WT: n = 281 cells; phosphomimetic: n = 128 cells).

---

cells activated CDK2 without CDK4/6 reactivation and progressed to the S phase within 48 h of drug treatment (Fig. 3d, e). These findings suggest that the reduction in Rb-protein levels alone is not sufficient for effective Rb inactivation and E2F activation. Thus, additional regulators may be required for augmenting E2F activity to initiate the cell cycle in the absence of CDK4/6 activity.

## c-Myc augments E2F activity after Rb-protein reduction
We next investigated the role of c-Myc in cell-cycle entry. Upon c-Myc knockdown, we observed a decrease in cyclin D1 expression and global mRNA transcription rate, as assessed by 5-ethynyl uridine (EU) incorporation (Fig. 4a–c). In addition, c-Myc knockdown led to a significant suppression of Rb phosphorylation at S807/811 in G0/G1, which was eliminated by acute CDK4/6i treatment for 15 min (Supplementary Fig. 4a, b). Conversely, c-Myc induction via a doxycycline-inducible system augmented both global mRNA transcription rate and Rb phosphorylation at S807/811 in G0/G1 (Fig. 4d and Supplementary Fig. 4c–f). These data suggest that c-Myc plays a pivotal role in modulating global transcriptional dynamics, subsequently influencing cyclin D expression and CDK4/6 activity.

Given the regulation of c-Myc by mitogenic signaling[29,30], we monitored ERK activity and cell proliferation in MCF-10A cells stably expressing ERK[31] and CDK2 reporters. Previous work established that variable ERK activity in mother cells guides CDK4/6 activation and

subsequent cell-cycle entry in daughter cells[27]. Aligning with these findings, our classification of ERK activity based on the CDK2 paths of daughter cells revealed higher ERK activity during the G2 phase of proliferating cells as opposed to quiescent cells (Supplementary Fig. 4g). Knockdown of c-Myc reduced cell proliferation in daughter cells and abolished the correlation between mother cell ERK activity and daughter cell proliferation (Supplementary Fig. 4h). As a positive control, similar effects were observed with cyclin D1/2/3 knockdown (Supplementary Fig. 4i), reinforcing the role of c-Myc as a principal mediator of mitogenic signaling for CDK4/6 activation in the canonical model of cell-cycle entry.

Using a doxycycline-inducible system, we increased c-Myc levels and performed live-cell imaging to evaluate alterations in cell-cycle entry after CDK4/6 inhibition. In parallel, we assessed the impact of cyclin D1 and cyclin E1 induction as canonical activators of CDK4/6 and CDK2, respectively. Our analysis revealed that cyclin E1 and c-Myc induction significantly elevated cell-cycle entry in the absence of CDK4/6 activity (Fig. 4e, f and Supplementary Fig. 5a–e). Despite mRNA induction of cyclin E1 and c-Myc, co-treatment with CDK4/6i and MEKi suppressed the increase in cell proliferation (Fig. 4f). These findings highlight the significance of mitogenic signaling and c-Myc in initiating CDK4/6-independent cell-cycle entry.

Our findings led us to postulate that c-Myc induction could enhance the modest E2F transcriptional activity resulting from the Rb-

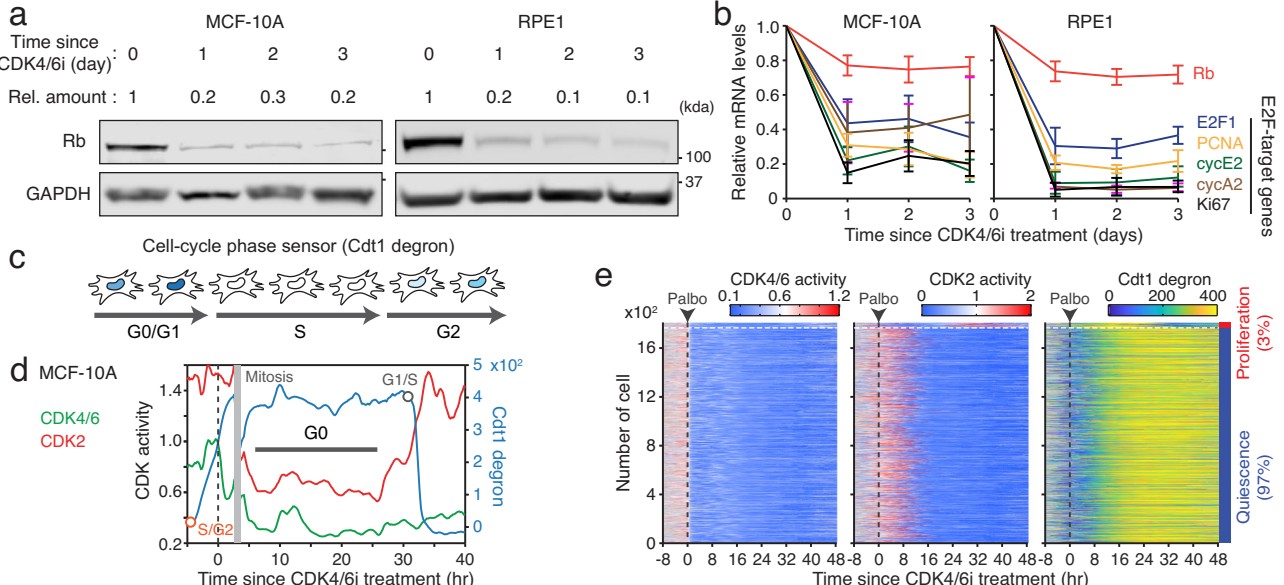

**Fig. 3 | Ineffective E2F activation by Rb-protein reduction. a** Immunoblot showing Rb and GAPDH expression in MCF-10A and RPE1 cells treated with palbociclib (1 μM). **b** Relative mRNA level changes in Rb and E2F-target genes in MCF-10A and RPE1 cells after treatment with palbociclib (1 μM). Data are shown as mean ± SEM ($n$ = 4 biological replicates). **c** Schematic diagram of the cell-cycle phase sensor based on Cdt1 degron. **d** Example of a single-cell trace showing CDK4/ 6 and CDK2 activities and Cdt1 degron levels after palbociclib (1 μM) treatment. Gray and orange circles mark the G1/S and S/G2 transitions, respectively. **e** Heatmap of single-cell traces for CDK4/6 and CDK2 activities and Cdt1 degron levels in MCF-10A cells treated with palbociclib (1 μM). During 25–48 h after drug treatment, cells activating CDK2 (>1) for over 2 h were classified as proliferating cells.

protein reduction, thus facilitating CDK2 activation and cell-cycle progression without CDK4/6 activity. To test this hypothesis, we examined the sequential impact of c-Myc induction on global transcription, Rb-protein levels, and E2F activity. We observed that c-Myc induction significantly increased the global transcription rate within 8 h (Fig. 4g and Supplementary Fig. 5f). However, CDK4/6 inhibition initially arrested cell proliferation and equally reduced Rb-protein levels and E2F activity within 12 h, regardless of c-Myc induction (Fig. 4h–j and Supplementary Fig. 5g–j). Notably, we found that c-Myc induction subsequently resulted in a significant increase in the percentage of S-phase cells and E2F activity after the initial response to CDK4/6 inhibition and Rb-protein reduction. These results indicate that c-Myc augments E2F activity following Rb-protein reduction in the absence of CDK4/6 activity.

**Competition between mitogenic and DNA damage signaling**

To identify additional potential regulators of CDK4/6-independent cell-cycle entry, we analyzed the top 100 genes that genetically interact with Rb using the Dependency Map[32]. Cyclin E1 deletion emerged as the leading candidate suppressing cell proliferation in Rb-knockout cancer cells (Fig. 5a), corroborating that CDK2 activation drives cell-cycle entry in the absence of CDK4/6 activity. Conversely, deletion of Cip/Kip family protein members, p21 (ranked #4) and p27 (ranked #2), along with the upstream p53 (ranked #3), were identified as top candidates facilitating Rb-independent cell-cycle progression. To assess the role of Cip/Kip family proteins in cell-cycle entry, we employed MCF-10A cells lacking expression of p21, p27, and p57 (triple knockout [tKO])[8]. We traced CDK4/6 and CDK2 activities and the level of Cdt1-degron and classified cells into proliferation and quiescence based on CDK2 activity. In the absence of CDK4/6 inhibition, nearly all tKO MCF-10A cells (99%) activated CDK4/6 and CDK2 for cell-cycle entry (Fig. 5b left and Supplementary Fig. 6a). Although CDK4/6i treatment still induced near-complete cell-cycle arrest and Rb-protein reduction within 24 h, the percentage of cells initiating CDK4/6-independent CDK2 activation was significantly higher in tKO cells than in wild-type cells (Fig. 5b right, c and Supplementary Fig. 6b). We next used EdU

incorporation for 48 h to determine the cumulative percentage of cells entering the cell cycle in the presence of CDK4/6i. To exclude CDK4/6-dependent cell proliferation, we first induced cell-cycle arrest via a 24-h CDK4/6i treatment before the EdU incubation. To access CDK4/6-dependent cell-cycle entry, we transiently induced EdU incorporation for 15 min. p53 knockout significantly facilitated both CDK4/6-independent and -dependent cell-cycle entry compared to wild-type cells (Supplementary Fig. 6c). Using the DHFR-TMP protein stabilization system, we found that p27 induction significantly suppressed both CDK4/6-dependent and -independent cell-cycle entry (Fig. 5d and Supplementary Fig. 6d). Moreover, increased p27 expression impeded cell proliferation in tKO MCF-10A cells stably entering the CDK4/6-independent cell cycle, established through chronic CDK4/6i exposure (Supplementary Fig. 6e). CDK4/6i treatment increased p21 and p27 levels, but not p57 levels (Supplementary Fig. 6f). These results demonstrate the involvement of the p53/p21 pathway and p27 protein to CDK4/6-independent cell proliferation.

We next employed MCF-7 cells, which express the CDK2 sensor and have p53 and p21 endogenously tagged with different fluorescent proteins[33], to examine the correlation between p53 and p21 levels and cell proliferation. Despite being breast cancer, MCF-7 cells retain an intact Rb/E2F pathway[34]. We classified cells based on the CDK2 paths and averaged time courses of p53 and p21 levels after treatment with either DMSO or CDK4/6i. Both protein levels were lower in cells entering both CDK4/6-dependent and -independent cell proliferation as compared to cells entering quiescence (Fig. 5e and Supplementary Fig. 6g). We also observed that levels of p53 were anti-correlated with CDK4/6-independent cell-cycle entry even before CDK4/6i treatment, suggesting that the history of p53-pathway activation determines p21 levels and subsequent cell proliferation (Fig. 5e). Furthermore, we found that co-treatment with CDK4/6i and MEKi significantly reduced CDK4/6-independent cell-cycle entry, c-Myc levels, and the global mRNA transcription rate in tKO MCF-10A cells (Supplementary Fig. 7). Together, our data imply that Cip/Kip family proteins regulate CDK4/6-independent cell-cycle entry after initial cell-cycle arrest, Rb-protein reduction, and upregulation of mitogenic signaling.

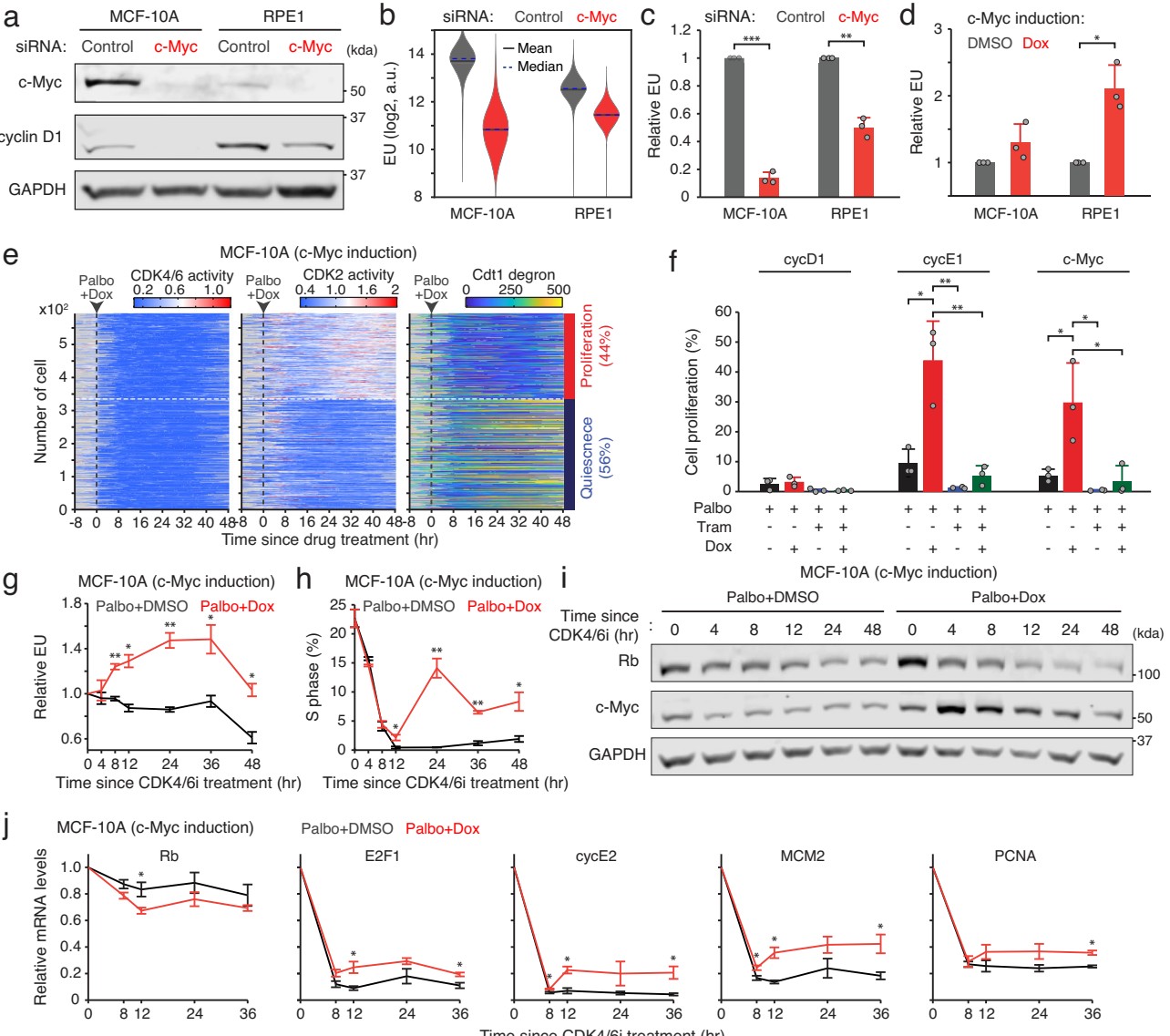

**Fig. 4 | Sequential regulation of E2F activity by Rb-protein reduction and c-Myc-mediated amplification. a** Immunoblot showing c-Myc, cyclin D1, and GAPDH expression 48 h after control or c-Myc siRNA knockdown. **b** Violin plots showing distribution of EU levels 48 h after control or c-Myc siRNA knockdown (*n* = 1500 cells/condition). **c, d** Relative EU levels 48 h after control or c-Myc siRNA knockdown (**c**) or 24 h after DMSO or doxycycline (5 μM) treatment (**d**). EU levels were normalized to the control condition. Data are shown as mean ± SD (*n* = 3 biological replicates). Asterisks indicate significant differences in the two-tailed unpaired *t*-test (*$p \le 0.05$; **$p \le 0.001$; ***$p \le 0.0001$). **e** Heatmap of single-cell traces for CDK4/6 and CDK2 activities and Cdt1 degron levels in MCF-10A cells treated with palbociclib (1 μM) + doxycycline (5 μM). During 25–48 h after drug treatment,

cells activating CDK2 (>1) for over 2 h were classified as proliferating cells. **f** Percentage of proliferating cells treated with indicated combinations of palbociclib (1 μM), trametinib (100 nM), and doxycycline (5 μM). Data are shown as mean ± SD (*n* = 3 biological replicates). Asterisks indicate significant differences in the one-way ANOVA test (*$p \le 0.05$; **$p \le 0.001$). **g–j** Relative EU levels (**g**), percentage of S-phase cells (**h**), immunoblot showing Rb, c-Myc, and GAPDH expression (**i**), and relative mRNA level changes in Rb and E2F-target genes (**j**). MCF-10A cells expressing a doxycycline-inducible c-Myc construct were treated with palbociclib (1 μM) + DMSO or doxycycline (5 μM). Data are shown as mean ± SEM (*n* = 3 biological replicates) (**g**, **h**, **j**). Asterisks indicate significant differences in the two-tailed unpaired *t*-test (*$p \le 0.05$; **$p \le 0.001$; ***$p \le 0.0001$).

Cip/Kip family proteins sense intrinsic and extrinsic stress, such as endogenous and exogenous DNA damage[35]. We next tested the impact of mitogenic and DNA damage signaling on the proliferation-quiescence decision in the absence of CDK4/6 activity. To assess CDK4/6-independent cell proliferation under different levels of mitogens and DNA damages, we incorporated long-term EdU (48 h) and titrated mitogens and neocarzinostatin (NCS) after halting cell proliferation with 24-h pre-treatment with CDK4/6i (Fig. 5f, g top). NCS treatment generates exogenous DNA double-stranded breaks. To increase the percentage of CDK4/6-independent cell-cycle entry, we used tKO cells for mitogen titration and wild-type cells with c-Myc induction for NCS titration. We found that mitogen and NCS titration

had opposing and graded effects on CDK4/6-independent cell proliferation (Fig. 5f, g bottom and Supplementary Fig. 8a, b). These results indicate that mitogenic and DNA damage signaling control CDK4/6-independent cell proliferation. The convergence of mitogenic and stress signaling on the same-cell fate decision raised the question of whether they competitively regulate cell-cycle entry without CDK4/6 activity. To test this hypothesis, we used wild-type MCF-10A cells stably entering the CDK4/6-independent cell cycle via chronic exposure to CDK4/6i for over a month. The half-maximal inhibitory concentration (IC50) of CDK4/6i was higher in drug-resistant cells than in drug-naïve cells (Supplementary Fig. 8c). We also confirmed cell proliferation in CDK4/6i-resistant cells

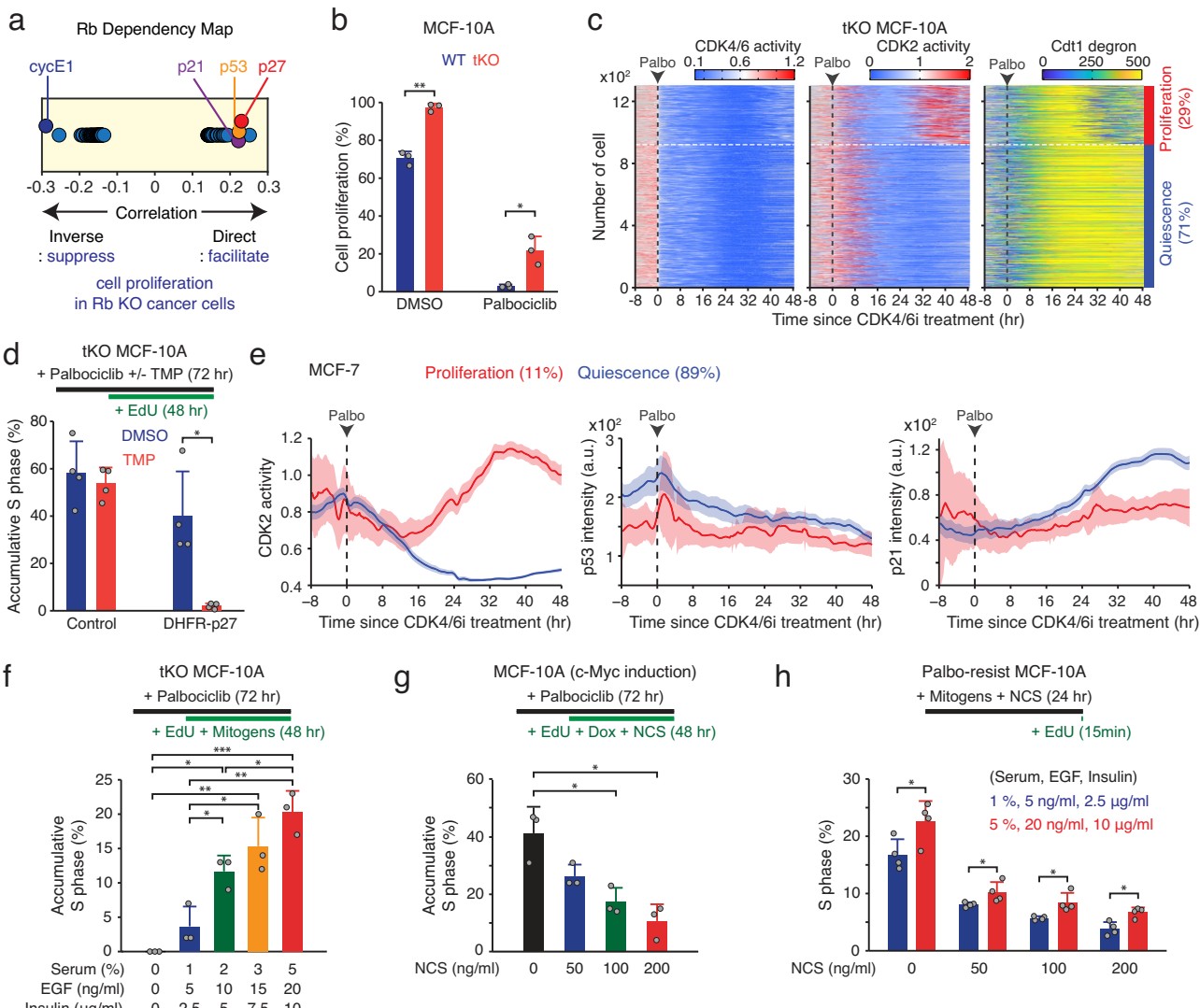

**Fig. 5 | Coordination of CDK4/6-independent cell proliferation through the competitive interplay between mitogenic and DNA damage signaling.**
**a** Dependency map of Rb from the Cancer Dependency Map Project[32]. **b** Percentage of proliferating cells in wild-type and tKO MCF-10A cells treated with either DMSO or palbociclib (1 μM). Data are shown as mean ± SD (n = 3 biological replicates). Asterisks indicate significant differences in the two-tailed unpaired t-test (*p ≤ 0.05; **p ≤ 0.001). **c** Heatmap of single-cell traces for CDK4/6 and CDK2 activities and Cdt1 degron levels in tKO MCF-10A cells treated with palbociclib (1 μM). During 25–48 h after drug treatment, cells activating CDK2 (>1) for over 2 h were classified as proliferating cells. **d** Percentage of S-phase cells in tKO MCF-10A cells without and with expression of a DHFR-p27 construct. Data are shown as mean ± SD (n = 4

biological replicates). Asterisks indicate significant differences in the two-tailed unpaired t-test (*p ≤ 0.05). **e** Average traces of CDK2 activity and p53 and p21 levels in MCF-7 cells treated with palbociclib (1 μM). Data are shown as mean ± 95% CI (proliferation: n = 447 cells; quiescence: n = 3582 cells). **f, g** Accumulative percentage of S-phase cells in tKO (**f**) or wild-type (**g**) MCF-10A cells. Data are shown as mean ± SD (n = 3 biological replicates). Asterisks indicate significant differences in the one-way ANOVA test (*p ≤ 0.05; **p ≤ 0.001; ***p ≤ 0.0001). **h** Percentage of S-phase cells in palbociclib-resistant MCF-10A cells exposed to different concentrations of mitogens and NCS as indicated for 24 h. Data are shown as mean ± SD (n = 4 biological replicates). Asterisks indicate significant differences in the two-tailed unpaired t-test (*p ≤ 0.05).

(Supplementary Fig. 8d). Furthermore, we observed a Rb-protein reduction (80%) but similar Rb-mRNA levels in CDK4/6i-resistant cells compared to drug-naïve cells (Supplementary Fig. 8e, f), suggesting that Rb-protein reduction is associated with CDK4/6i resistance. We found that higher levels of mitogens significantly reversed the effect of NCS-mediated DNA damage on CDK4/6-independent cell proliferation (Fig. 5h). These data suggest that mitogenic and DNA damage signaling competitively control the proliferation-quiescence decision in the absence of CDK4/6 activity.

**Targeting mitogenic signaling in cancer**
Most driver mutations in cancer occur in mitogenic signaling pathways, implying that cancer cells may express high levels of c-Myc and more likely adapt to CDK4/6 inhibition than non-transformed cells. To

test this, we used two uveal melanoma cell lines (MP41 and MP46) harboring G-protein mutations but expressing intact Rb. Uveal melanoma cell lines exhibited higher levels of phosphorylated ERK (p-ERK) and AKT (p-AKT) as well as c-Myc levels compared to non-transformed MCF-10A and RPE1 cells (Supplementary Fig. 9a–c). We next compared the responses and adaptation to CDK4/6 inhibition between non-transformed and uveal melanoma cells. Emerging evidence suggests that rapidly drug-adapting cancer cells (persisters) drive residual cancer growth and the development of drug-resistant populations[36,37]. CDK4/6i treatment initially induced near-complete suppression of cell proliferation in both non-transformed and melanoma cells within 1 day (Fig. 6a and Supplementary Fig. 9d–g). However, we found that a higher percentage of melanoma cells (1.1–7.1%) developed persisters and entered the S phase within 3 days of CDK4/6 inhibition compared

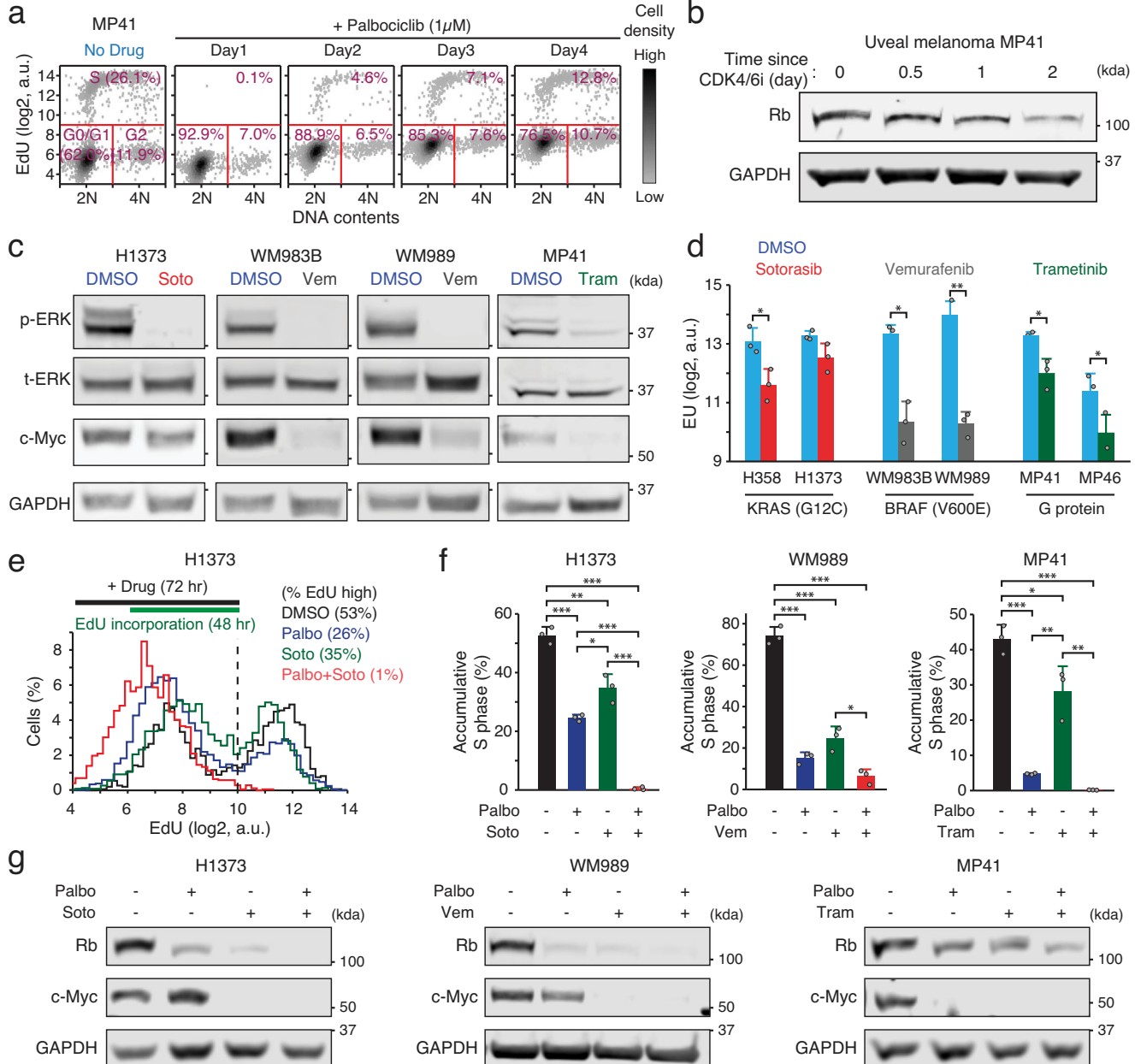

**Fig. 6 | Synergistic inhibition of drug-tolerant persister cells through co-targeting mitogenic signaling and CDK4/6 activity. a** Density scatterplot of Hoechst and EdU staining in MP41 cells treated with palbociclib (1 μM) for the indicated time (*n* = 1000 cells/condition). **b** Immunoblot showing Rb and GAPDH expression in MP41 cells after palbociclib (1 μM) treatment. **c** Immunoblot showing p-ERK, t-ERK, c-Myc, and GAPDH expression in H1373, WM983B, WM989, and MP41 cells treated with DMSO or the indicated drug for 24 h (sotorasib, 1 μM; vemurafenib, 1 μM; trametinib, 10 nM). **d** EU levels 24 h after treatment with the indicated drug. Data are shown as mean ± SD (*n* = 3 biological replicates). Asterisks indicate significant differences in the two-tailed unpaired *t*-test (*$p \leq 0.05$; **$p \leq 0.001$). **e** Representative histograms of EdU levels in H1373 cells showing classification of S-phase cells. 24 h after treatment with the indicated drug (palbociclib, 1 μM; sotorasib, 1 μM), cells were incubated with EdU (10 μM) for 48 h prior to fixation (*n* > 3000 cells/condition). **f** Percentage of S-phase cells in H1373, WM989, and MP41 cells. Data are shown as mean ± SD (*n* = 3 biological replicates). Asterisks indicate significant differences in the one-way ANOVA test (*$p \leq 0.05$; **$p \leq 0.001$; ***$p \leq 0.0001$). **g** Immunoblot showing Rb, c-Myc, and GAPDH expression in H1373, WM989, and MP41 cells 72 h after treatment with the indicated drug.

to non-transformed cells (0–1.5%). Moreover, CDK4/6i treatment in MP41 cells led to noticeable Rb-protein reduction before the emergence of persisters (Fig. 6a, b), suggesting the contribution of Rb-protein reduction to CDK4/6i adaptation in cancer cells. Our data indicate that upregulated mitogenic signaling in cancer may cause more effective adaptation to CDK4/6i than non-transformed cells.

We examined whether targeting mitogenic signaling in cancer suppresses c-Myc-mediated transcriptional activity and consequently persisters. To target upregulated mitogenic signaling in cancer, we used the KRAS[G12C] inhibitor sotorasib, the BRAF inhibitor

vemurafenib, and trametinib in *KRAS[G12C]*-mutant non-small cell lung cancer (H358 and H1373), *BRAF[V600E]*-mutant melanoma (WM983B and WM989), and G protein-mutant uveal melanoma (MP41 and MP46) cell lines, respectively. Targeting activating mutations in mitogenic signaling or MEK activity effectively reduced p-ERK and c-Myc levels as well as global mRNA transcription rate (Fig. 6c, d and Supplementary Fig. 9h, i). To evaluate the percentage of persisters, we next used EdU incorporation for 48 h and excluded CDK4/6-dependent cell proliferation by pre-treating cells with CDK4/6i for 24 h (Fig. 6e top). Targeting mitogenic signaling in combination with CDK4/6i

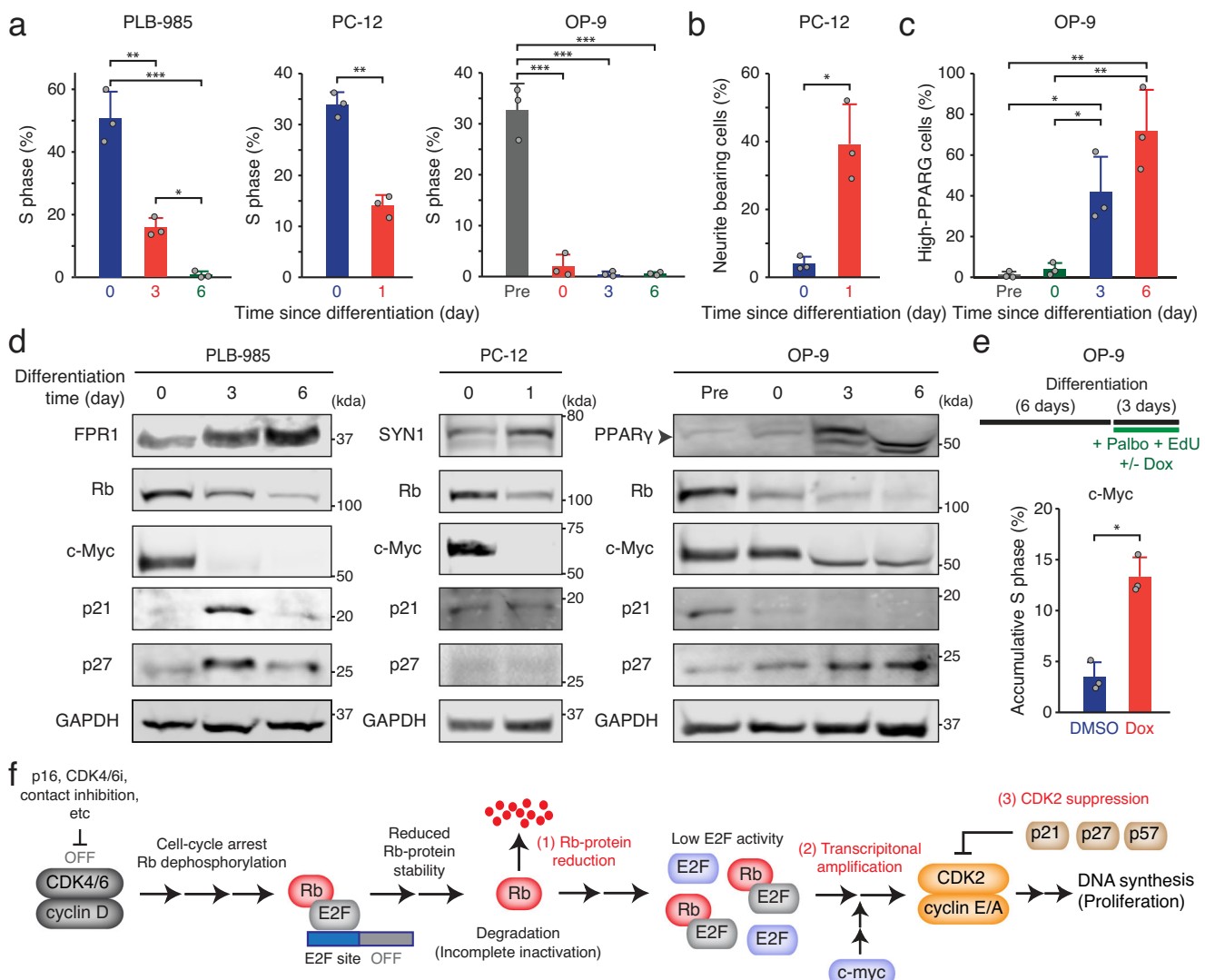

**Fig. 7 | Control of c-myc and Cip/Kip levels for the maintenance of quiescence in differentiated cells. a** Percentage of S-phase cells in PLB-985, PC-12, and OP-9 cells after differentiation for the indicated time. Data are shown as mean ± SD (*n* = 3 biological replicates). Asterisks indicate significant differences in the two-tailed unpaired *t*-test (PC-12 cells) or one-way ANOVA test (PLB-985 and OP-9 cells) (*$p \leq 0.05$; **$p \leq 0.001$; ***$p \leq 0.0001$). **b** Percentage of neurite bearing PC-12 cells before and after one-day differentiation. Data are shown as mean ± SD (*n* = 3 biological replicates). Asterisks indicate significant differences in the two-tailed unpaired *t*-test (*$p \leq 0.05$). **c** Percentage of high-PPARγ expressing OP-9 cells. Data are shown as mean ± SD (*n* = 3 biological replicates). Asterisks indicate significant

differences in the one-way ANOVA test (*$p \leq 0.05$; **$p \leq 0.001$). **d** Immunoblot showing Rb, c-Myc, p21, p27, and GAPDH expression before and after differentiation. FPR1, SYN1, and PPARγ are cell differentiation markers. **e** Percentage of S-phase OP-9 cells expressing a doxycycline-inducible c-Myc construct. After 6 days of differentiation, cells were treated with palbociclib (1 μM) and EdU (10 μM) + DMSO or doxycycline (5 μM) for 72 h. Data are shown as mean ± SD (*n* = 3 biological replicates). Asterisks indicate significant differences in the two-tailed unpaired *t*-test (*$p \leq 0.05$). **f** Schematic diagram illustrating CDK4/6-independent cell-cycle entry by multiple steps: (1) reduction in Rb-protein levels, (2) c-Myc-mediated amplification of E2F activity, and (3) inhibition of CDK2 activity by Cip/Kip.

synergistically inhibited the emergence of persisters despite the most noticeable loss of Rb protein (Fig. 6e–g). These results indicate that upregulated mitogenic signaling in cancer facilitates CDK4/6i adaptation via c-Myc-mediated enhancement of transcriptional activity after Rb-protein reduction.

**Maintenance of quiescence in differentiated cells**
Although cellular quiescence refers cell-cycle arrest that can be reversed depending on intrinsic and extrinsic factors, it has been traditionally believed that cell proliferation in terminally differentiated cells is irreversibly halted[38]. Rb protein is a fundamental mediator of cell differentiation and lineage specification[39]. Therefore, we next investigated whether differentiation processes regulate the expression levels of Rb protein, c-Myc, and Cip/Kip family proteins, thereby

governing cell proliferation. We used three in vitro cell differentiation models: neutrophil-like PLB-985, neuron-like PC-12, and adipocyte-like OP-9 cells. The differentiation process of these cells induced cell-cycle arrest (Fig. 7a) and increased percentages of PC12 cells bearing neurite-like elongation (Fig. 7b and Supplementary Fig. 10a) and OP-9 cells expressing PPARγ, the master regulator of adipocyte differentiation (Fig. 7c and Supplementary Fig. 10b, c). We observed a noticeable reduction in Rb protein as cells entered quiescence and underwent differentiation processes, which were confirmed by increased levels of differentiation markers, FPR1, SYN1, and PPARγ (Fig. 7d). Moreover, we observed downregulation of c-Myc and upregulation of p21 and/or p27 during the differentiation processes. These results suggest that despite loss of Rb protein, differentiated cells regulate the expression levels of c-Myc and Cip/Kip to maintain the quiescent state.

To investigate the reversibility of quiescence, we conducted experiments to determine whether c-Myc induction could reverse the quiescent state and trigger CDK4/6-independent cell proliferation in differentiated cells. After differentiating OP-9 cells into adipocyte-like stage for 6 days, we used a doxycycline-inducible system and EdU incorporation for 3 days to increase c-Myc levels and assess the accumulative percentage of S-phase entry, respectively (Fig. 7e top). In addition, we treated cells with CDK4/6i to ensure CDK4/6-independent cell proliferation. Our results showed that c-Myc induction in differentiated OP-9 cells significantly increased the percentage of cells entering S phase in the presence of CDK4/6i (Fig. 7e bottom). Furthermore, c-Myc induction reduced the percentage of OP-9 cells expressing PPARγ (Supplementary Fig. 10d). These findings suggest that differentiated cells have the capacity to re-enter cell proliferation.

## Discussion

Our study provides mechanistic insights into the molecular mechanism underlying cell-cycle entry in the absence of CDK4/6 activity. While the canonical pathway regulating cell-cycle entry is well-characterized, the alternative model thorough which mammalian cells bypass CDK4/6 activity and initiate Rb inactivation remained elusive. Our data indicate that the non-phosphorylated form of Rb is unstable, leading to its reduction at the protein level. By modulating c-Myc and Cip/Kip levels together with CDK4/6 inhibition, we observed that an initial cell-cycle arrest and Rb downregulation preceded CDK2 activation and S-phase entry. These results suggest a sequential order of these key players in governing CDK4/6-independent cell-cycle entry. Based on these results, we propose that CDK4/6-independent cell-cycle entry is coordinated by multiple sequential steps: (1) reduction of Rb-protein levels, (2) c-Myc-mediated amplification of E2F activity, and (3) regulation of Cip/Kip levels (Fig. 7f).

In addition to Rb inactivation by phosphorylation, our data suggest an alternative pathway for Rb inactivation by reducing its levels. However, in contrast to effective Rb inactivation by active CDK4/6, this alternative pathway of Rb inactivation is passive, incomplete, and less effective. It may only lead to minimal and leaky E2F activation. Given the slow kinetics of non-phosphorylated Rb-protein reduction ($t_{1/2} = 4$ h), our data argue that non-phosphorylated Rb is unstable and subsequently leads to its reduction. Therefore, Rb-protein levels are dependent on the cell-cycle status in unperturbed cycling cells and reach a low steady state approximately 20 h after entering quiescence. Furthermore, as cells re-enter the cell cycle, Rb phosphorylation stabilizes and increases its levels. This regulation of Rb-protein levels may delay E2F activity and cell-cycle re-entry, providing a safety mechanism that ensures favorable conditions for cell-cycle re-entry. We also note that previous studies described Rb degradation mediated by E3 ubiquitin ligases, including NRBE3[40], TRIM28[41], and β-TRCP1[40], suggesting the possible involvement of the E3-ubiquitin proteasome pathway in the regulation of Rb-protein levels. NRBE3 is an E2F1 target gene, and TRIM28 binds phosphorylated Rb[40,41], inducing degradation of phosphorylated Rb and reinforcing CDK4/6-dependent cell proliferation. Although overexpression of β-TRCP1 does not induce Rb-protein reduction[41], it promotes Rb-protein degradation after CDK4/6i treatment[42]. Nevertheless, other studies failed to detect Rb ubiquitination[43–45] but found that MDM2[43] and the protein TTC39B[44] promote degradation of Rb protein in a ubiquitination-independent and proteasome-dependent manner.

Our data show that c-Myc mediates mitogenic signaling to induce cyclin D expression and CDK4/6 activation, facilitating canonical cell-cycle entry. Moreover, when CDK4/6 activity is inhibited, c-Myc expression is required to amplify low E2F transcriptional activity after Rb-protein reduction. Thus, cells entering quiescence due to low mitogenic signaling still reduce Rb levels but may not enter the CDK4/6-independent cell cycle. Given that most driver mutations in cancer upregulate mitogenic signaling, which stabilizes c-Myc expression,

cancer cells are likely more resistant to CDK4/6i than non-transformed cells. Aggressive cancer cells expressing high c-Myc levels may readily initiate cell proliferation without CDK4/6 activity and show primary resistance to CDK4/6i. Although CDK4/6i therapy is successful in HR⁺ breast cancer when combined with hormone therapy, which suppresses c-Myc expression[46], CDK4/6i monotherapy often insufficiently blocks cancer progression due to the development of drug resistance. Our data show that targeting cancer-specific mutations in mitogenic signaling effectively reduced c-Myc and mRNA transcription rate, suppressing the development of CDK4/6i-tolerant persisters in cancer. These results are consistent with previous results showing that targeting either ERK or PI3K signaling in combination with CDK4/6i synergistically suppresses the emergence of drug resistance[47–49]. In addition, most mutations driving CDK4/6i resistance in patients with HR⁺ breast cancer occur in mitogenic signaling genes[50]. However, unexpected toxicities with combinations of CDK4/6 and PI3K or mTOR inhibitors severely limit clinical success[11]. Thus, targeting cancer-specific mutations in mitogenic signaling in combination with CDK4/6i may be an optimal strategy to minimize toxicity while maximizing their therapeutic outcomes. Our data also indicate that despite near-complete loss of Rb protein, differentiated cells downregulate c-Myc levels to maintain quiescence. These results are supported by previous studies showing that c-Myc is one of the Yamanaka factors that play a vital role in the creation of induced pluripotent stem cells from differentiated cells[51]. Moreover, previous reports also showed that terminally differentiated skeletal myotubes can re-enter the cell cycle upon growth factor stimulation, indicating that cell-cycle arrest in differentiated cells is reversible[52,53].

Intrinsic and extrinsic stress, such as DNA damage and contact inhibition, induces expression of Cip/Kip family proteins[35]. Thus, after initiating E2F and CDK2 activation without CDK4/6 activity, upregulation of Cip/Kip family proteins blocks cell-cycle entry. This provides another safety mechanism to ensure that both intrinsic and extrinsic conditions are favorable for cell proliferation. Consistently, a recent report showed that temporal application of cytotoxic chemotherapy followed by CDK4/6i treatment synergistically suppresses the growth of pancreatic adenocarcinoma[54]. In addition, loss-of-function mutations of *p53* are associated with worse therapeutic outcomes of CDK4/6i treatment in HR⁺ breast cancer[55]. These findings also indicate that Cip/Kip family proteins play an important role in the development of CDK4/6i resistance in cancer cells.

Together, our study outlines a series of sequential steps involved in the CDK4/6-independent cell-cycle entry, offering insights into the mechanisms by which mammalian cells initiate proliferation under CDK4/6-limited conditions and sustain extended quiescence. Although the significance of this alternate pathway in normal cell physiology remains uncertain, our findings could have potential implications for advancing cancer therapeutic approaches.

## Methods
### Cell culture
MCF-10A cells (ATCC, #CRL-10317) were cultured in DMEM/F12 medium (Gibco, #11039047) supplemented with 5% horse serum (HS) (Gibco, #16050122), 20 ng/ml EGF (Peprotech, # HZ-1326), 10 μg/ml insulin (Sigma, #I1882), 500 μg/ml hydrocortisone (Sigma, #H0888), and 100 ng/ml cholera toxin (Sigma, #C8052). RPE1 cells (ATCC, #CRL-4000) were maintained in DMEM/F12 medium containing 10% fetal bovine serum (FBS) (Gibco, #10437028) and 0.01 mg/ml hygromycin B (InvivoGen, #ant-hg). HS68 (ATCC, #CRL-1635), MCF-7 (ATCC, #HTB-22), and H358 cells (ATCC, #CRL-5807) were grown in DMEM medium (Genesee Scientific, #25-500) with 10% FBS supplementation. H1373 (ATCC, #CRL-5866) and PLB-985 cells were cultured in RPMI-1640 medium (Genesee Scientific, #25-506) containing 10% FBS. PLB-985 cells, a subclone of the HL-60 cell line, were acquired from Dr. Orion Weiner's laboratory. MCF-7 cells expressing p53 and p21 endogenously

tagged with different fluorescent proteins were obtained from Dr. Galit Lahav's laboratory. The PLB-985 cell line has been identified as a misidentified subclone of the HL-60 cell line in the ICLAC database. While we have not verified the authenticity of our PLB-985 cultures, these cells are recognized models for human neutrophils. MP41 (ATCC, #CRL-3297) and MP46 (ATCC, #CRL-3298) cells were maintained in RPMI-1640 medium supplemented with 20% FBS. WM989 (Rockland Immunochemicals, #WM989-01-0001) and WM983B cells (Rockland Immunochemicals, #WM983B-01-0001) were cultured in a mixture of 80% MCDB153 (Sigma, #M7403) and 20% Leibovitz's L15 media (Gibco, #21083027), supplemented with 2% FBS and 1.68 mM CaCl₂. PC-12 cells (ATCC, #CRL-1721) were grown in RPMI-1640 medium containing 5% FBS and 10% HS on collagen-prehybridized plates (30 μg/ml) (Advanced BioMatrix, #5005-B). OP-9 cells (ATCC, #CRL-2749) were maintained in MEM-α medium (Gibco, #12571048) supplemented with 20% FBS on poly-D-lysine-prehybridized plates (20 μg/ml) (Sigma, #A-003-E).

For mitogen starvation experiments, MCF-10A cells were washed with three times with PBS and cultured in DMEM/F12 medium containing 0.3% bovine serum albumin (BSA), 500 μg/ml hydrocortisone, and 100 ng/ml cholera toxin. Other cell lines were cultured in their respective growth media without FBS. All cells were maintained at 37 °C with 5% $CO_2$ and tested negative for mycoplasma. In live-cell microscopy experiments, phenol red-free media were used to minimize background fluorescence.

### Cell differentiation
PLB-985 cells were differentiated into a neutrophil-like phenotype by adding 1.3% DMSO (Sigma, #D2438) to the culture media[56]. PC-12 cells were induced to differentiate using RPMI-1640 medium supplemented with 0.5% FBS, 1.0% HS, and 50 ng/mL rat beta-NGF (R&D System, #556NG100)[57]. OP-9 cells were differentiated by culturing them to confluence and then incubating them in MEM-α media containing 15% Knockout Serum (Gibco, #10828028), which is characterized by a high insulin concentration[58].

### Live-cell reporters
In this study, we utilized live-cell reporters to monitor individual cells and detect CDKs activities and cell-cycle phase transitions. Histone 2B (H2B) was used to track individual cells, while fluorescently-tagged Cdt1 degron (a.a. 1–100)[28] was employed to identify cell-phase transitions. The Cdt1 degron undergoes rapid degradation during the S phase and reaccumulates at the beginning of G2, thereby detecting G1/S and S/G2 transitions.

Previously described KTRs for CDK4/6 and CDK2 were also employed in this study[25,26]. These reporters are comprised of a fluorescently tagged substrate specific to the kinase of interest, conjugated to a peptide containing a bipartite nuclear localization sequence (NLS) and a nuclear export sequence (NES). A fragment of Rb C-terminus (a.a. 886–928) and a fragment of human DNA helicase B (DHB) (a.a. 994–1087) were selected as substrate for CDK4/6 and CDK2, respectively. Kinase-specific phosphorylation determines the competition between NLS and NES, and the ratio of fluorescence in the cytoplasm versus the nucleus serves as a readout for kinase activity.

The CDK2 reporter measures the collective activity of cyclin E/A-CDK1/2 complexes[59]. Considering that KTRs contain a degenerate CDK2 substrate motif from the CDK2 sensor, the CDK4/6 sensor detects signals measured by the CDK2 sensor during S and G2 phases[26,60]. Therefore, as described previously[26], we applied a correction factor calculated using linear regression in MCF-10A cells. By subtracting a calculated fraction of the CDK2 reporter signal from the CDK4/6 reporter signal, the CDK4/6 activity was determined as follows in both wild-type and tKO MCF-10A cells (Supplementary Fig. 11): CDK4/6 activity = CDK4/6 reporter activity − 0.35 × CDK2 reporter activity.

### Plasmid generation
Previous described constructs include pLenti-DHB (a.a. 995–1087)-mVenus-p2a-mCherry-Rb (a.a. 886–928)-IRES-blasticidin (CDK2 and CDK4/6 sensors[26]: Addgene, #126679), pPBbsr2-EKAR-NLS (ERK FRET sensor[31]), pLenti-YFP-Rb-IRES-blasticidin[45], and pLenti-YFP-Rb (All Ser/Thr to Glu)-IRES-puromycin[45]. Gibson cloning was employed to insert iRFP670-H2B, p2a-mCerulean, and a fragment of Cdt1 (a.a. 1–100) into pLenti-IRES-puromycin. The nuclear marker construct was generated by incorporating H2B-iRFP670 into pLenti-IRES-neomycin. NLS-mRuby3 was generated through PCR amplification of the mRuby3 gene along with synthetic oligo-DNAs encoding the NLS sequence derived from SV40 (PKKKRKV) and inserted into pLenti-IRES-puromycin vector by Gibson cloning. Doxycycline-inducible constructs were created by inserting PCR products of cyclin D1, cyclin E1, and c-Myc into the lentiviral pCW57.1 vector (Addgene, #50661) following NheI and BamHI restriction digestion. DHFR fused to p16 or p27 was cloned into pCru5-IRES-puromycin.

### Cell line generation
Stable cell lines were generated by introducing constructs into cells using either lentiviral or retroviral transduction. For the lentiviral transduction, HEK-293T cells were co-transfected with lentiviral plasmids, which included the cDNA of interest, along with pMDLg/pRRE (Addgene, #12251), pRSV-rev (Addgene, #12253), and pCMV-VSV-G (Addgene, #8454) using PEI transfection reagent. In the retroviral transduction, HEK-293T cells were co-transfected with retroviral plasmids containing the target cDNA and pCL-Ampho (Novus Biologicals, #NBP2-29541), also using the PEI reagent. 72 and 96 h post-transfection, the virus-containing supernatant was harvested. It underwent a 5-min centrifugation, was filtered through a 0.45 μM filter unit (Millipore, #SLHA033SB), and then concentrated with an ultra-centrifugal filter (Millipore, #UFC910024). These concentrated viruses were preserved at −80 °C until required.

For the creation of cells with multiple constructs, different antibiotic selection markers were embedded in each plasmid. Plasmids were then introduced into cells one by one. Following transduction, cells were either selected with specific antibiotics (10 μg/ml puromycin for RPE1 and 1 μg/ml for other cell lines, 10 μg/ml blasticidin, or 200 μg/ml neomycin) or were sorted with an Influx cell sorter based on the fluorescence of the reporter, two days after transduction.

Previously described cell lines include p21/p27/p57 tKO MCF-10A[8], MCF-7 expressing fluorescently-tagged p21 and p53[33], and MCF-10A cells expressing ERK and CDK2 sensors[27]. To establish CDK4/6i resistance, MCF-10A cells were exposed to palbociclib (1 μM) for over a month. Palbociclib resistance was confirmed by measuring IC50 and the percentage of S-phase cells.

### Drugs and chemicals
Stock solutions of the following drugs were prepared by dissolving them in DMSO (Sigma, #D2438): palbociclib (Selleck Chemicals, #S1116), trametinib (Selleck Chemicals, #S2673), vemurafinib (Selleck Chemicals, #S1267), sotorasib (MedChemExpress, #HY-114277), doxycycline (Sigma, #D9891), and trimethoprim (Cayman Chemical, #16473). Cycloheximide (Sigma, #C7698) and neocarzinostatin (Sigma, #N9162) were procured as ready-to-use solution. 5-Ethynyl-2'-Deoxyuridine (EdU) was obtained from Sigma (#900584). 5-Ethynyl-Uridine (EU) was purchased from Click Chemistry Tools (#1261).

### Antibodies
The following antibodies were purchased from Cell Signaling Technology: phospho-Rb (Ser807/811) (#8516; 1:2000 for immunofluorescence), Rb (#9309; 1:2000 for immunofluorescence and immunoblotting), c-Myc (#5605; 1:1000 for immunofluorescence), phospho-ERK1/2 (Thr202/Tyr204) (#4370; 1:3000 for immunoblotting), ERK1/2 (#4696; 1:3000 for immunoblotting), phospho-AKT

(Ser473) (#4060; 1:2000 for immunoblotting), AKT (#2920; 1:2000 for immunoblotting), PPARγ (#2435; 1:100 for immunofluorescence; 1:1000 for immunoblotting), p21 (#2947; 1:2000 for immunoblotting in PLB-985 cells), p27 (#3686; 1:1600 for immunofluorescence; 1:2000 for immunoblotting in PLB-985 and PC-12 cells), p57 (#2557; 1:250 for immunofluorescence), and GAPDH (#2118; 1:5000 for immunoblotting). Anti-cyclin D1 (#MA5-14512; 1:500 for immunoblotting) antibody was obtained from Thermo Scientific. Anti-FPR1 (#391602; 1:1000 for immunoblotting) antibody was procured from BioLegend. Anti-SYN1 (#A17362; 1:1000 for immunoblotting) antibody was sourced from ABclonal. Antibodies against Rb (#ab181616; 1:2000 for immunoblotting in mouse and rat cell lines) and c-Myc (#ab32072; 1:1000 for immunoblotting) were purchased from Abcam. Anti-p21 (#556430; 1:500 for immunofluorescence) and anti-p27 (#610241; 1:2000 for immunoblotting in OP-9 cells) antibodies were acquired from BD Biosciences. Anti-p21 (#sc-271610; 1:300 for immunoblotting in PC-12 and OP-9 cells) was ordered from Santa Cruz Biotechnology. Alexa Fluor 488 goat anti-rabbit (#A32731), Alexa Fluor 568 goat anti-rabbit (#A11036), Alexa Fluor 488 goat anti-mouse (#A32723), and Alexa Fluor 568 goat anti-mouse secondary antibodies (#A11031) were obtained from Thermo Scientific. Anti-rabbit HRP-linked antibody (#7074) was from Cell Signaling Technology. IRDye 800CW goat anti-mouse (#926-32210) and IRDye 680RD goat anti-rabbit (#926-68071) secondary antibodies were purchased from LI-COR Biosciences. All secondary antibodies were used at a 1:2000 dilution in their respective blocking solutions.

## siRNA transfection

c-Myc targeting siRNA duplexes (1: 5′-AUCAUUGAGCCAAAU-CUUAAAAAAA-3′; 2: 5′-GACGAGACCUUCAUCAAAAACAUC-3′) and a non-targeting negative control siRNA were procured from Integrated DNA Technologies and used at a concentration of 20 nM. Transfections were performed using DharmaFECT1 (Horizon, #T-2001) following the manufacturer's instructions. Briefly, cells were seeded at least one night before transfection. DharmaFECT1 and 2 µM siRNA stock were diluted 1:10 (final 20 nM) in Opti-MEM medium (Gibco, #31985070) and incubated at room temperature for 10 min. The diluted DharmaFECT1 and siRNA were then combined and incubated at room temperature for 10–20 min. The siRNA/DharmaFECT1 mixture was added to the growth medium at a final composition of 20% mixture and 80% of growth media (transfection medium). Subsequently, the cultured cells' medium was replaced with the transfection medium for 48 h before fixation for immunofluorescence or harvesting for immunoblotting.

## Quantitative real-time polymerase chain reaction

Total RNA was isolated using the GeneJET RNA Purification Kit (Thermo Scientific, #K0731) and reversed-transcribed using the qScript™ cDNA SuperMix (VWR, #101414-106) according to the manufacturers' protocols. Real-time PCR was conducted using the PerfeCTa® SYBR® Green FastMix® (VWR, #101414-286) on a Quant-Studio 6 Flex real-time PCR system (Applied Biosystems, #4485691). The reaction was initiated with a hot-start $Taq$ DNA polymerase activation step at 95 °C for 2 min, followed by 40 cycles of denaturation at 95 °C for 5 s, and annealing and extension at 60 °C for 30 s. A melting curve stage was run at 95 °C for 15 s, 60 °C for 1 min, and 95 °C for 15 s. All reactions were performed in triplicate. Relative mRNA levels were calculated using the $2^{-\Delta\Delta Ct}$ method by normalizing the expression to a housekeeping gene, RSP23. The primers for the real-time PCR reactions were procured from Integrated DNA Technologies and were as follows: $Rb1$: 5′-CTCTCGTCAGGCTTGAGTTTG-3′ (forward), 5′-GAC ATCTCATCTAGGTCAACTGC-3′ (reverse); $E2F1$: 5′-TCTCGGCCAGG-TACTGATG-3′ (forward), 5′-ACCCTGACCTGCTGCTCTT-3′ (reverse); $Cyclin E1$, 5′-TCTTTGTCAGGTGTGGGGA-3′ (forward), 5′-GAAATGGC-CAAAATCGACAG-3′ (reverse); $Cyclin E2$: 5′-TCAAGACGAAGTAGCCG

TTTAC-3′ (forward), 5′-TGACATCCTGGGTAGTTTTCCTC-3′ (reverse); $Cyclin A2$: 5′-TGGAAAGCAAACAGTAAACAGCC-3′ (forward), 5′-GGG CATCTTCACGCTCTATTT-3′ (reverse); PCNA: 5′-CCTGCTGGGA-TATTAGCTCCA-3′ (forward), 5′-CAGCGGTAGGTGTCGAAGC-3′ (reverse); $MCM2$: 5′-CCGTGACCTTCCACCATTTGA-3′ (forward), 5′-GGTAGTCCCTTTCCATGCCAT-3′ (reverse); $Ki67$: 5′-ACGCCTGGT-TACTATCAAAAGG-3′ (forward), 5′-CAGACCCATTTACTTGTGTTGGA-3′ (reverse); $RSP23$: 5′-TTCTTGATCAGCTGGACCCT-3′ (forward), 5′-ACCCTTTTGGAGGTGCTTCT-3′ (reverse).

## Immunoblotting

Whole cell lysates were collected using CHAPS lysis buffer (50 mM Tris-HCl, 150 mM NaCl, 1 mM EDTA, 10 mM NEM, and 0.3% CHAPS in deionized water) containing halt protease inhibitor cocktail (Thermo Scientific, #1861279) and phosphatase inhibitor (Sigma, #4906845001), unless otherwise stated. The lysates were obtained after ice-cold PBS washes and cleared by 10 min of centrifugation at $14,000 \times g$ at 4 °C. OP-9 cells were collected with RIPA lysis buffer (Sigma, #R0278) to measure c-Myc, and with CHAPS lysis buffer to measure other proteins. Proteins from this cell line were further purified by a methanol purification step. Briefly, they were precipitated from the lysis buffer with 90% (v/v) methanol and pelleted by 10 min of centrifugation at $14,000 \times g$ at room temperature. The pellets were washed twice with 90% methanol, air dried, and re-dissolved in the lysis buffer. Protein concentrations were measured using the Pierce™ 660 nm protein assay reagent (Thermo Scientific, #22660) following the manufacturer's instructions. The proteins were mixed with LDS sample buffer (Invitrogen, #NP0007) containing 2% 2-mercaptoethanol and denatured at 70 °C for 10 min. They were resolved on a 1.0 mm 4–12% Bis-Tris gel (Invitrogen) and transferred to a PDVF membrane using a power blotter-semi-dry transfer system (Bio-Rad Trans-Blot Turbo system, #1704150) according to the manufacturer's instructions. The membranes were incubated in the blocking solution (LI-COR, #927-60001) for 1 h at room temperature and then incubated with primary antibodies diluted in the blocking solution overnight at 4 °C. The next day, after 5 TBST washes, membranes were incubated with either goat anti-mouse IR Dye 800CW (LI-COR, #926-32210) or goat anti-rabbit IR Dye 680RD (LI-COR, #926-68071) secondary antibody at a dilution of 1:2000 in blocking solution for 1 h. Blots were scanned with the LI-COR Odyssey Infrared Imaging System, and analysis was conducted using Image Studio Lite software (LI-COR, version 3.1.).

In case where fluorescence-based detection did not provide a strong signal, proteins were detected using chemiluminescence (p27 in PC-12). Membranes were incubated with goat anti-rabbit immunoglobulin−horseradish peroxidase conjugates (Cell Signaling Technology, #7074) at a dilution of 1:2000 in blocking solution for 1 h and then reacted with the chemiluminescent substrate (Sigma, #WBLUF0100). Proteins were detected using the ChemiDoc system (Bio-Rad, version 5.2.1.). For membrane reblotting, membranes were incubated in the stripping buffer (Thermo Scientific, #46430) for 15 min at room temperature and then treated as in the first round, starting with blocking.

## Immunofluorescence

Fixed-cell experiments were carried out using 96-well glass-bottomed plates (Cellvis, #P96-1.5P). Cells were seeded into wells at least a night before the experiments. For experiments involving EdU or EU staining, cells were incubated with EdU (10 µM) or EU (1 mM) for 15 min at 37 °C before fixation, unless specified otherwise. Cells were fixed with 2% paraformaldehyde (PFA) for 15 min at room temperature. This was achieved by adding a 1:1 ratio of 4% PFA in PBS (Thermo Scientific, #28906) containing 10 mM HEPES to the culture medium. The cells were then permeabilized with 0.2% triton X-100 in PBS for 15 min. EdU or EU labeling was performed using a Click-IT EdU or EU detection kit

with AFDye 647 picolyl azide (Click Chemistry Tools, #1300), following the manufacturer's guidelines. Cells were incubated in blocking buffer (PBS containing 0.1% triton X-100, 10% FBS, 1% BSA, and 0.01% $NaN_3$) for 1 h at room temperature. Subsequently, cells were stained overnight at 4 °C with primary antibodies diluted in blocking buffer. The next day, Alexa Fluor 488- or 568-conjugated secondary antibodies were applied at a 1:2000 dilution in the blocking buffer for 1 h at room temperature. Nuclei were stained with Hoechst 33342 dye (Thermo Scientific, #62249) (10 mg/ml) diluted in PBS (1:2000) for 15 min at room temperature before imaging. After each step, cells were washed with PBS and stored in PBS for imaging.

### RNA FISH
The RNA FISH procedure was performed using the ViewRNA ISH Cell Assay Kit (Thermo Scientific, #QVC0001), with custom probes targeting ACTB, Rb, E2F1, and CDC25A purchased from Invitrogen. Briefly, cells were cultured on 96-well plates pre-treated with collagen (30 μg/ml) (Advanced BioMatrix, #5005-B) for at least 1 h. Cells fixation and permeabilization were carried out as described in the immunofluorescence section. Probe hybridization, amplification, and Alexa Fluor 555 labeling were conducted following the manufacturer's protocols, prior to EdU and primary antibody staining.

### Fixed-cell and live-cell imaging
Fluorescence fixed-cell images were captured using a Nikon microscope, mounted onto an inverted Eclipse Ti-2 body using 20× objective (Nikon CFI Plan Apo Lambda, 0.75 NA, 2-by-2-pixel binning). Thirty-two nonoverlapping sites were imaged per well. For live-cell imaging, 96-well plates were placed in a humidified 37 °C chamber (Tokai) with 5% $CO_2$, and images were taken with the Nikon microscope using either a 20× or 10× (Nikon CFI Plan Apo Lambda, 0.45 NA, no binning) objective. Three or four nonoverlapping sites in each well were imaged every 12 min per frame. Total light exposure time was kept below 600 ms for each time point. Acquisitions settings were adjusted to ensure an optimal signal-to-noise ratio without saturating the signal.

### Image processing and analysis
We performed automated image analysis using a custom MATLAB pipeline (MathWorks, R2021a). After flat-field correction for light bias, nuclei within cells were segmented and tracked. For fixed-cell analysis, Hoechst staining was employed to segment nuclei based on a threshold defined by histogram curvature. During live-cell analysis, the H2B-iRFP670 was automatically segmented to identify nuclei using a Laplacian of Gaussian blob detection method. For multi-round fixed-cell imaging, each imaging round was segmented and then aligned with the previous round to track individual cells.

To track cell movement in live-cell imaging, we utilized the deflection-bridging algorithm. Mitotic occurrences in live-cell imaging were pinpointed by spotting two proximate daughter nuclei in relation to a prior nucleus, ensuring their collective nuclear signal mirrored the preceding nucleus's signal. To assess the activities of KTR-based CDK2 and CDK4/6 reporters, the cytoplasmic region was spatially approximated by creating a ring with a 2 μm inner radius and a 10 μm outer radius from the nuclear mask. The cytoplasmic rings intersected with neighboring cell cytoplasmic rings were omitted. The signal of each fluorescent channel was calculated after global background subtraction. The background signal was estimated as the median intensity of all non-masked pixels after excluding the nuclear mask dilated by 50 μm. Nuclear signal was measured based on the nuclear segmentation, and cytoplasmic signal was measured within a ring outside of the nucleus (0.65 μm to 3.25 μm outside the edge of the nucleus). All immunofluorescence signals were calculated as the medium nuclear or cytoplasmic intensity unless otherwise noted. For RNA FISH analysis, we estimated the entire cell region by formulating an area that

enveloped the nucleus and extended up to 50 μm from the nucleus mask without overlaps with adjacent cells. A top hat filter with a 4 μm radius circular kernel was employed on raw images to produce the FISH puncta mask. We manually selected different intensity thresholds for each probe and cell line to minimize false positives and negatives. Regions surpassing these thresholds were quantified.

### Statistics and reproducibility
Statistical analyses were performed using GraphPad Prism v.10.1.0. For parametric data, we utilized unpaired two-tailed Student's $t$-tests or one-way ANOVA with Tukey's post hoc tests, as appropriate, to assess pairwise differences. When comparing two-paired groups, paired two-tailed Student's $t$-tests were employed. Statistical test results are reported in the figure legends and detailed in Supplementary Data 1. All experiments were conducted in at least two independent experiments.

### Reporting summary
Further information on research design is available in the Nature Portfolio Reporting Summary linked to this article.

## Data availability
The raw image data are available under restricted access due to large file size. Access can be obtained upon request to the corresponding author: Hee Won Yang (hy2602@cumc.columbia.edu). No original code is reported in this paper. Source data are provided with this paper.

## Code availability
Imaging analysis code is available at https://github.com/tjdtn5160/image-analysis-kim-2023.

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

## Acknowledgements

We thank K. Aoki and M. Matsuda for EKAR sensors, J. Stewart-Ornstein and G. Lahav for the p21 and p53 tagged MCF-7 cell line, O. Weiner for PLB-985 cell line, and the members of the Yang lab and M. Kim for helpful comments and suggestions. This work was supported by Research Scholar Grant (H.Y., RSG-22-101-01-CDP) from the American Cancer Society, V Scholar Grant (H.Y., V2023-017) from the V foundation, Melanoma Research Foundation Grant (H.Y.), and R01 Grant (H.Y., R01-GM145884) from the NIH. These studies used the resources of the Herbert Irving Comprehensive Cancer Center Flow Cytometry Shared Resources funded in part through Center Grant (P30-CA013696).

## Author contributions

Conceptualization, experimental design, investigation, interpretation, and visualization: M.Z., S.K., and H.Y.; writing: S.K and H.Y.; supervision and funding acquisition: H.Y.

## Competing interests

The authors declare no competing interests.
