## [Peer Review File · Nature Communications]

Non-canonical pathway for Rb inactivation and external signaling coordinate cell-cycle entry without CDK4/6 activityREVIEWER COMMENTS

Reviewer #1 (Remarks to the Author):

This is a difficult-to-summarize manuscript as it touches many different ideas using a large number of techniques, reporters and different cell lines. Figures 1-3 are basically repetition of variation of previous data in which cell cycle entry in cells with defective CDK4/6 activity mostly depends on other cyclin-dependent, CIP/KIP-modulated activities. Figure 4 and 5 reports the relevance of cyclin E and MYC in driving proliferation in the absence of CDK4/6 activity. Figure 6 investigates the relative relevance of mitogenic signaling and DNA damage. Figure 7 suggests that MYC is important downstream of other oncogenic signals in melanoma cells, and Fig 8 reports the relevance of MYC and CIP/KIP proteins during differentiation in neutrophil-like, neuron-like and adipocyte-like cells. Lastly, Figure 8 illustrates how exogenous MYC is able to induce cell cycle entry in differentiated cells irrespective of CDK4/6 activity.

In general, most of these assays are confirmatory of previous reports suggesting the relevance of RB1 degradation, the role of cyclin E-related activity and CIP/KIP proteins in cell cycle progression after inhibition of CDK4/6, the balance between mitogenic signals and DNA damage in cell cycle entry etc. The use of so many cellular scenarios, or different cell types with diverse mitogenic/oncogenic backgrounds, etc, makes it difficult to digest the information. It may be my fault, but I could not generate a clear picture of what the message of the paper is. I don't understand the "sequentiality" suggested in the title, or the novelty in any of the sentences of the abstract. I do agree that the technical quality of the manuscript is high, but I am afraid that many readers may be as confused as I am.

Reviewer #2 (Remarks to the Author):

According to the canonical cell cycle entry model, the retinoblastoma protein Rb is inactivated through hyperphosphorylation by cyclin-Cdk complexes. While there is consensus regarding the significance of Rb phosphorylation, there is ongoing debate within the cell cycle field regarding the dynamics and distinct functions of G1 and S-phase cyclin-dependent complexes. In this manuscript, the authors provide evidence that inhibiting Cdk4/6 leads to Rb destabilization. Furthermore, they propose that under specific conditions, the interplay between Cdk4/6 inhibition, resulting in Rb degradation, along with mitogenic signaling and stress signaling, enhances E2F activity and therefore regulates cell cycle entry.

I think this manuscript offers valuable contributions to the field by 1) introducing an alternative pathway for Rb inactivation and 2) providing insights into the regulation of cell cycle entry in the absence of Cdk4/6 activity. Consequently, I consider this manuscript to meet the standards for publication in the journal Nature Communications. However, it is crucial to note that improvements are necessary before I can provide my full support. Enhancements should be made to address the following concerns:

Major points:

-The authors' comprehensive effort, employing four different methods to inhibit Cdk4/6 activity, is certainly impressive. However, it raises the question of whether the observed effects are a direct result of Cdk4/6 inhibition that solely affects Rb or if there is a global effect through indirect activation of other components, such as the proteasome. This possibility is mentioned in a previous study with Palbociclib (PMID: 29669860) and should be addressed in this work.

-To critically test their model, it would be valuable for the authors to demonstrate that Cdk4/6 inhibition effectively eliminates Rb phosphorylation, leading to the degradation of Rb. Authors could accomplish this by examining the stability of non-phosphorylatable Rb, such as Rb Δ Cdk as used in (PMID: 24876129). If the non-phosphorylatable Rb exhibits increased instability similar to the exogenous Rb after Cdk4/6 inhibition (as shown in Figure 2), it would provide further support for their proposed mechanism.

Minor:

-I appreciate the authors for their efforts in conducting quantitative western blots. Given that the Rb signals measured in the case of Cdk4/6 inhibition are relatively weak, it would be valuable for the authors to demonstrate that their measurements fall within a linear range. This will ensure the accuracy and reliability of their quantification. By performing a linearity analysis, such as using a serial dilution of the samples, the authors can provide evidence that their western blot measurements are indeed quantitative. This will enhance the overall quality of the manuscript as western blots are used throughout the manuscript.

-The authors' use a Rb C-terminus-based live cell reporter to measure Cdk4/6 activity in both wild-type and triple knockout (tKO) MCF-10A cells. However, it would be beneficial for the authors to provide an explanation regarding the reasoning behind using the same correlation factor (0.35) to calculate the final Cdk4/6 activity for both cell backgrounds. This information would help readers understand why the same factor is applicable in different cellular contexts and provide insights into the consistency and validity of the approach across different Cdk activity backgrounds.

-In addition to demonstrating the percentage of cells in the S phase of cell cycle in Extended Data Figure 2d and e, it would be valuable for the authors to quantify other cell cycle phases, including the G1 phase. This is particularly important considering that the exogenous Rb is expected to have the most significant impact during this phase.

-Line 87 should read contact inhibition instead of contract inhibition

-Line 222 should read cell proliferation instead of cell prolifearction

-Line 226 should read palbociclib instead of pabociclib

-Line 282 should read CDK4/6-independent instead of CDK4/6-indpendent

-Line 427 should read CDK4/6 instead of CD4/6

Response to Reviewers:

We would like to express our sincere appreciation to the reviewers for their insightful comments and suggestions. We have diligently implemented the recommended changes in our revised manuscript. Below, we provide a detailed account of the modifications made in response to each reviewer's comments as well as the original critiques. For the convenience of the reviewers, we have highlighted all changes in the manuscript.

We are confident that the revisions, made in alignment with the reviewers' valuable suggestions, have considerably enhanced the quality of our manuscript. Furthermore, we have included a summary table of the newly added figures for ease of reference. It is our aspiration that we have adequately addressed all the concerns and queries raised by the reviewers and that this revised manuscript now meets the publication standards of *Nature Communications*.

Newly added figures	Description
Supplementary Fig. 1g, h	Quantification of Rb-protein serial dilutions.
Fig. 2d	Stability of Rb protein in cells entering quiescence.
Fig. 2f	Stability of Rb protein in cells treated with DMSO or CDK4/6i.
Fig. 2g	Stability of wild-type or phosphomimetic Rb protein in the presence of CDK4/6i.
Supplementary Fig. 2g, h	Relative changes in YFP-Rb, H2B-iRFP, and NLS-mRuby3 after treatment with CDK4/6i.
Supplementary Fig. 11	Calibration of non-specific signals in the CDK4/6 reporter during S/G2 phases based on the CDK2 reporter.

REVIEWER COMMENTS

Reviewer #1 (Remarks to the Author):

This is a difficult-to-summarize manuscript as it touches many different ideas using a large number of techniques, reporters and different cell lines. Figures 1-3 are basically repetition of variation of previous data in which cell cycle entry in cells with defective CDK4/6 activity mostly depends on other cyclin-dependent, CIP/KIP-modulated activities. Figure 4 and 5 reports the relevance of cyclin E and MYC in driving proliferation in the absence of CDK4/6 activity. Figure 6 investigates the relative relevance of mitogenic signaling and DNA damage. Figure 7 suggests that MYC is important downstream of other oncogenic signals in melanoma cells, and Fig 8 reports the relevance of MYC and CIP/KIP proteins during differentiation in neutrophil-like, neuron-like and adipocyte-like cells. Lastly, Figure 8 illustrates how exogenous MYC is able to induce cell cycle entry in differentiated cells irrespective of CDK4/6 activity.

In general, most of these assays are confirmatory of previous reports suggesting the relevance of RB1 degradation, the role of cyclin E-related activity and CIP/KIP proteins in cell cycle progression after inhibition of CDK4/6, the balance between mitogenic signals and DNA damage in cell cycle entry etc. The use of so many cellular scenarios, or different cell types with diverse mitogenic/oncogenic backgrounds, etc, makes it difficult to digest the information. It may be my fault, but I could not generate a clear picture of what the message of the paper is. I don't understand the "sequentiality" suggested in the title, or the novelty in any of the sentences of the abstract. I do agree that the technical quality of the manuscript is high, but I am afraid that many readers may be as confused as I am.

Response: We appreciate the reviewer's valuable feedback. To increase the accessibility and better explain the novelty of our study, we have undertaken a comprehensive revision of our manuscript.

1. To underscore our identification of an alternative pathway for Rb inactivation and highlight the influence of external signaling, we have modified the title of our study, as follows:

"Non-canonical pathway for Rb inactivation and external signaling sequentially coordinate cell-cycle entry without CDK4/6 activity"

2. We have restructured our abstract to illuminate the unique aspects of this study. The standard mechanism of cell-cycle entry, involving CDK4/6-mediated Rb phosphorylation, is well-documented. However, the pathways through which mammalian cells can enter the cell cycle without CDK4/6 activity, and the mechanisms driving this alternative Rb inactivation, have remained unclear. Furthermore, the interplay between this alternative Rb inactivation and external signaling was not well understood.

3. To enhance the readability of our manuscript, we have made our focus and central message more apparent. We've thoroughly restructured our figures and text. All amendments in the manuscript are highlighted for ease of reference.

4. Our data suggest a sequential regulation of E2F activation and cell-cycle entry that operates independently of CDK4/6. **Figures 1 and 2** show that cells entering quiescence by CDK4/6 inhibition dramatically reduce expression levels of Rb protein regardless of its expression levels and promoter activity. **Figure 3** illustrates that despite a significant reduction in Rb-protein, cells maintain low E2F activity when CDK4/6 is inhibited. This indicates that the decrease in Rb-protein leads to partial inactivation of Rb, resulting in insufficient E2F activity. Hence, it proposes the need for other factors to enhance E2F activity. Consistent with this notion, **Figure 4** demonstrates that following the reduction in Rb-protein, the introduction of c-Myc expression augments E2F activity, thereby facilitating cell-cycle entry without relying on CDK4/6 activity. Further, **Figure 5** shows that proteins of the Cip/Kip family can inhibit cell-cycle entry independent of CDK4/6 after the reduction in Rb-protein and activation of E2F and CDK2. **Figures 6 and 7** indicate that suppressing c-Myc expression or

induction of Cip/Kip family proteins in cancer and differentiated cells can cause cell-cycle arrest despite the loss of Rb protein. Collectively, our data implies a sequential regulation of cell-cycle entry that does not require CDK4/6 activity, as illustrated in **Figure 7f**.

Reviewer #2 (Remarks to the Author):

According to the canonical cell cycle entry model, the retinoblastoma protein Rb is inactivated through hyperphosphorylation by cyclin-Cdk complexes. While there is consensus regarding the significance of Rb phosphorylation, there is ongoing debate within the cell cycle field regarding the dynamics and distinct functions of G1 and S-phase cyclin-dependent complexes. In this manuscript, the authors provide evidence that inhibiting Cdk4/6 leads to Rb destabilization. Furthermore, they propose that under specific conditions, the interplay between Cdk4/6 inhibition, resulting in Rb degradation, along with mitogenic signaling and stress signaling, enhances E2F activity and therefore regulates cell cycle entry.

I think this manuscript offers valuable contributions to the field by 1) introducing an alternative pathway for Rb inactivation and 2) providing insights into the regulation of cell cycle entry in the absence of Cdk4/6 activity. Consequently, I consider this manuscript to meet the standards for publication in the journal Nature Communications. However, it is crucial to note that improvements are necessary before I can provide my full support. Enhancements should be made to address the following concerns:

Major points:

-The authors' comprehensive effort, employing four different methods to inhibit Cdk4/6 activity, is certainly impressive. However, it raises the question of whether the observed effects are a direct result of Cdk4/6 inhibition that solely affects Rb or if there is a global effect through indirect activation of other components, such as the proteasome. This possibility is mentioned in a previous study with Palbociclib (PMID: 29669860) and should be addressed in this work.

Response: We appreciate the reviewer for raising an important question. To address this, we monitored YFP-Rb together with mRuby3-nuclear localization signal (NLS) and histone 2B (H2B)-iRFP. If the Rb-protein reduction is mediated by a global effect of other components, such as proteasome activity, CDK4/6 inhibition may also reduce NLS-mRuby3 and H2B-iRFP intensities. We found that CDK4/6i treatment only reduced YFP-Rb, but neither NLS-mRuby3 nor H2B-iRFP. These data suggest that the Rb-protein reduction is not due to increased proteasome activity. These new data are displayed in **Supplementary Fig. 2g, h** below.

Supplementary Figure 2.

g, h Heatmap of single-cell traces for H2B-iRFP670, YFP-Rb, and NLS-mRuby3 levels in MCF-10A cells treated with DMSO (**g**) or palbociclib (1 μ M) (**h**). Each row represents a single-cell trace over time according to the respective color map.

-To critically test their model, it would be valuable for the authors to demonstrate that Cdk4/6 inhibition effectively eliminates Rb phosphorylation, leading to the degradation of Rb. Authors could accomplish this by examining the stability of non-phosphorylatable Rb, such as Rb Δ Cdk as used in (PMID: 24876129). If the non-phosphorylatable Rb exhibits increased instability similar to the exogenous Rb after Cdk4/6 inhibition (as shown in Figure 2), it would provide further support for their proposed mechanism.

Response: We appreciate the reviewer's insightful comment and suggestion. In response, we carried out the recommended experiments. To test whether the stability of Rb protein is regulated by Rb phosphorylation status, we generated a phosphomimetic Rb construct, replacing 14 CDK phosphorylation sites (Serine/Threonine) with glutamic acid. Following CDK4/6 inhibition for 24 hr to remove phosphorylation in wild-type Rb, cells expressing either wild-type or phosphomimetic Rb were treated with cycloheximide to block protein synthesis. We found that the half-life of the phosphomimetic Rb protein was about five times longer than that of the wild-type Rb protein (**Fig. 2g**). These new results indicate that the non-phosphorylated state of Rb is unstable, reducing Rb-protein levels in quiescent cells. We have incorporated these data in **Fig. 2g**, as presented below.

Figure 2.

g Time course of relative YFP tagged wild-type and phosphomimetic Rb in MCF-10A cells treated with cycloheximide (10 $\mu\text{g/ml}$). Cells were treated with palbociclib (1 μM) for 24 hr before imaging (WT: $n = 281$ cells; phosphomimetic: $n = 128$ cells).

Minor:

-I appreciate the authors for their efforts in conducting quantitative western blots. Given that the Rb signals measured in the case of Cdk4/6 inhibition are relatively weak, it would be valuable for the authors to demonstrate that their measurements fall within a linear range. This will ensure the accuracy and reliability of their quantification. By performing a linearity analysis, such as using a serial dilution of the samples, the authors can provide evidence that their western blot measurements are indeed quantitative. This will enhance the overall quality of the manuscript as western blots are used throughout the manuscript.

Response: We appreciate the reviewer for raising the important question to increase the quality of our protein quantification. We conducted the recommended experiment and performed a linearity analysis of Rb-protein quantification. We have incorporated these data in **Supplementary Fig. 1g** and **h** below.

Supplementary Figure 1.

g Immunoblot showing Rb levels across various amounts of total lysates in MCF-10A and RPE1 cells.

h Relative Rb-protein levels. Solid lines represent the best fitted lines.

-The authors' use a Rb C-terminus-based live cell reporter to measure Cdk4/6 activity in both wild-type and triple knockout (tKO) MCF-10A cells. However, it would be beneficial for the authors to provide an explanation regarding the reasoning behind using the same correlation factor (0.35) to calculate the final Cdk4/6 activity for both cell backgrounds. This information would help readers understand why the same factor is

applicable in different cellular contexts and provide insights into the consistency and validity of the approach across different Cdk activity backgrounds.

Response: We appreciate the reviewer's question. Considering that KTRs contain a degenerate CDK2 substrate motif from the CDK2 sensor, the CDK4/6 sensor detects signals measured by the CDK2 sensor during S and G2 phases¹⁻³. Therefore, as described previously², we applied a correction factor calculated using linear regression in MCF-10A cells. To explain the application of the correction factor in both wild-type and tKO MCF-10A cells, we have included an additional figure (**Supplementary Fig. 11**) as shown below.

Supplementary Figure 11. Calibration of non-specific signals in the CDK4/6 reporter during S/G2 phases based on the CDK2 reporter.

a, b, Heatmap of single-cell traces for uncorrected and corrected CDK4/6 activity and CDK2 activity sorted by the time of mitosis in wild-type (a) and tKO (b) MCF-10A cells. Cells were treated with palbociclib (1 μ M).

-In addition to demonstrating the percentage of cells in the S phase of cell cycle in Extended Data Figure 2d and e, it would be valuable for the authors to quantify other cell cycle phases, including the G1 phase. This is particularly important considering that the exogenous Rb is expected to have the most significant impact during this phase.

Response: To address this important issue, we quantified other cell cycle phases and incorporated this information into **Supplementary Fig. 2b-d** as outlined below.

Supplementary Figure 2.

b, c Representative density scatterplots showing Hoechst and EdU staining in MCF-10A (b) and RPE1 (c) cells without and with YFP-Rb expression ($n = 2,000$ cells/condition).

d Percentage of each cell-cycle phase in MCF-10A and RPE1 cells without and with YFP-Rb expression. Data are shown as mean \pm SD ($n = 3$ biological replicates).

-Line 87 should read contact inhibition instead of contract inhibition

- Line 222 should read cell proliferation instead of cell prolifeartion
- Line 226 should read palbociclib instead of pabociclib
- Line 282 should read CDK4/6-independent instead of CDK4/6-indpendent
- Line 427 should read CDK4/6 instead of CD4/6

Response: We are very grateful to the reviewer for their thorough reading of our manuscript and for pointing out these errors.

References for Response to Reviewer Comments:

- 1 Gerosa, L. *et al.* Receptor-Driven ERK Pulses Reconfigure MAPK Signaling and Enable Persistence of Drug-Adapted BRAF-Mutant Melanoma Cells. *Cell Syst* **11**, 478-494 e479 (2020).
<https://doi.org/10.1016/j.cels.2020.10.002>
- 2 Yang, H. W. *et al.* Stress-mediated exit to quiescence restricted by increasing persistence in CDK4/6 activation. *Elife* **9** (2020). <https://doi.org/10.7554/eLife.44571>
- 3 Kim, S., Carvajal, R., Kim, M. & Yang, H. W. Kinetics of RTK activation determine ERK reactivation and resistance to dual BRAF/MEK inhibition in melanoma. *Cell Rep* **42**, 112570 (2023).
<https://doi.org/10.1016/j.celrep.2023.112570>

REVIEWER COMMENTS

Reviewer #1 (Remarks to the Author):

The manuscript has improved the readability but I don't see any change in the problems detected in the previous revision.

a) In breast cancer cells, proteasome-dependent degradation of RB1 upon inhibition of CDK4/6 (Fig. 2; novel data), although data also indicates that this is insufficient to promote G1-S transition (Fig. 3), as expected given the efficacy of these inhibitors.

b) In breast cancer cells, Myc controls transcription and proliferation (Fig. 4a-c,g). Myc and cyclin E can induce cell cycle entry in the absence of CDK4/6 activity (Fig. 4e,f). Conclusions (lines 204-205): "Myc as a principal mediator of mitogenic signaling for CDK4/6 activation in the canonical model of cell-cycle entry"; (213-214) "significance of mitogenic signaling and c-Myc in initiating CDK4/6-independent cell-cycle entry": these two conclusions are already known. The third conclusion: (226-227) "sequential regulation of CDK4/6-independent E2F activation by RB-protein reduction and c-Myc mediated amplification of transcriptional activity": two separate events reported: the first one in a); the second one using an overexpression model (already known) with no data about sequentiality or endogenous conditions (I don't think the authors have supported that the effect of Myc "following the reduction of Rb-protein" (answers to the reviewers) requires reduced RB1 levels.

c) In breast cancer cells, CIP/KIP proteins prevent CDK4/6-dependent and -independent cell cycle entry. They also participate in DNA damage signaling, which counteracts cell cycle entry (Fig. 5). Conclusion (lines 296-298): "mitogenic and DNA damage signaling competitively control the proliferation-quiescence decision in the absence of CDK4/6 activity". Already known. (the authors indicated that these inhibitors can function "after the reduction of RB1 protein levels" but, again, no work to demonstrate that these inhibitors require reduced RB1 to do their function.

d) In melanoma cells, inhibiting major oncogenes (MEK-ERK pathway) prevents transcription and proliferation, and cooperates with CDK4/6 inhibition. Already known. The link with Myc is correlative.

e) In neutrophils, neurons and adipocytes, Myc is downregulated and CIP/KIP are upregulated. Already known. Suppressing Myc and overexpression of the inhibitors can suppress proliferation despite the loss of RB1.

192-193. “Myc regulates global transcriptional activity for CDK4/6 activation”. This conclusion comes from correlative data.

271-272: “Cip/Kip family proteins regulate CDK4/6-independent cell-cycle entry after E2F and CDK2 activation triggered by Rb-protein reduction and c-Myc-mediated amplification of transcriptional activity”. “by” makes no sense here.. authors are assuming from the beginning that should be the final conclusion of the paper.

In general, I am afraid that the manuscript confirms the relevance of several players whose function was already known and proposes a “sequential” effect and a “requirement” for RB1 reduction and Myc activation. Whereas the reduction of RB1 is a interesting observation, it has no functional consequences by itself. All the other assays overexpressing Myc or Cip/Kip etc. are confirmatory of previous data and lack physiological (or pathological) justification as presented.

Reviewer #2 (Remarks to the Author):

I thank authors for responding to my questions raised during the initial review of their paper. While the authors have been able to make their story more accessible for general reader and improve the manuscript during revision, I still have some additional comments and questions about the new data added. I think these points need answers before this manuscript meets the standards of Nature Communications.

Supplementary Figure 2b – In the representative density scatterplot for MCF-10A with Exo-Rb the total % extends over 100% (G0/G1 55%+S 33%+G2 16%). This quantification should be corrected in 2b and calculations changed in 2d.

Supplementary Figure 2d – Y-axis should be renamed as it shows now different cell cycle phases instead of S phase.

Supplementary Figure 2g and 2h – Can authors explain why there is a small but consistent transition around 13hr point consistently in all heatmaps showing single cell traces for H2B-iRFP, YFP-Rb and NLS-mRuby3.

Response to Reviewers:

We would like to express our sincere appreciation to the reviewers for their constructive suggestions. In response, we have incorporated the recommended modifications into our updated manuscript. Below, we outline the specific changes made based on each reviewer's feedback, accompanied by the original comments.

We believe that the revisions, made in alignment with the reviewers' valuable suggestions, have considerably enhanced the quality of our manuscript. Additionally, we have included a summary table of the revised figures for ease of reference. We hope that our revisions comprehensively address the reviewers' concerns and that this revised manuscript now meets the publication standards of *Nature Communications*.

Revised figures	Description
Supplementary Fig. 2g, h	Relative changes in YFP-Rb, H2B-iRFP, and NLS-mRuby3 after treatment with CDK4/6i.

REVIEWER COMMENTS

Reviewer #1 (Remarks to the Author):

The manuscript has improved the readability but I don't see any change in the problems detected in the previous revision.

a) In breast cancer cells, proteasome-dependent degradation of RB1 upon inhibition of CDK4/6 (Fig. 2; novel data), although data also indicates that this is insufficient to promote G1-S transition (Fig. 3), as expected given the efficacy of these inhibitors.

Response to a: We appreciate the reviewer's insightful feedback. Our temporal analysis shows that c-Myc and Cip/Kip proteins influence CDK4/6-independent cell-cycle progression, but this effect becomes evident only after the Rb-protein reduction in non-transformed cells. Consequently, our data underscore the potential for Rb-protein reduction to activate E2F activity in the absence of CDK4/6 activity.

b) In breast cancer cells, Myc controls transcription and proliferation (Fig. 4a-c,g). Myc and cyclin E can induce cell cycle entry in the absence of CDK4/6 activity (Fig. 4e,f). Conclusions (lines 204-205): "Myc as a principal mediator of mitogenic signaling for CDK4/6 activation in the canonical model of cell-cycle entry"; (213-214) "significance of mitogenic signaling and c-Myc in initiating CDK4/6-independent cell-cycle entry": these two conclusions are already known. The third conclusion: (226-227) "sequential regulation of CDK4/6-independent E2F activation by RB-protein reduction and c-Myc mediated amplification of transcriptional activity": two separate events reported: the first one in a); the second one using an overexpression model (already known) with no data about sequentiality or endogenous conditions (I don't think the authors have supported that the effect of Myc "following the reduction of Rb-protein" (answers to the reviewers) requires reduced RB1 levels.

c) In breast cancer cells, CIP/KIP proteins prevent CDK4/6-dependent and -independent cell cycle entry. They also participate in DNA damage signaling, which counteracts cell cycle entry (Fig. 5). Conclusion (lines 296-298): "mitogenic and DNA damage signaling competitively control the proliferation-quiescence decision in the absence of CDK4/6 activity". Already known. (the authors indicated that these inhibitors can function "after the reduction of RB1 protein levels" but, again, no work to demonstrate that these inhibitors require reduced RB1 to do their function.

Response to b and c: We appreciate the insights from the reviewer. Our findings demonstrate that, even with c-Myc induction, CDK4/6 inhibition leads to cell-cycle arrest and a decrease in Rb-protein levels, despite an increase in overall transcriptional activity (Fig. 4g-i and Supplementary Fig. 5f-h). Moreover, inducing c-Myc appears to enhance E2F activity subsequent to the reduction of Rb-protein levels (Fig. 4j and Supplementary Fig. 5i, j). These observations point towards a sequential regulation of E2F activation: first through the reduction of Rb-protein and then amplified by c-Myc's effect on transcriptional activity.

d) In melanoma cells, inhibiting major oncogenes (MEK-ERK pathway) prevents transcription and proliferation, and cooperates with CDK4/6 inhibition. Already known. The link with Myc is correlative. 192-193. "Myc regulates global transcriptional activity for CDK4/6 activation". This conclusion comes from correlative data.

Response to d: We appreciate the reviewer for highlighting this question. Our findings demonstrate that a knockdown of c-Myc led to a decrease in both the CDK4/6 activator cyclin D and overall transcriptional activity (Fig. 4a-c). On the other hand, inducing c-Myc amplified global transcriptional activity (Fig. 4d and Supplementary Fig. 4c, d). Furthermore, changes in c-Myc levels, through knockdown or induction, correspondingly influenced Rb phosphorylation in the G0/G1 phase, which can be near completely eliminated

by acute treatment with CDK4/6i for 15 min (Supplementary Fig. 4a, b, e, f). These results indicate the direct relationship between c-Myc and CDK4/6 activity. To provide clarity on this matter, we have made modifications to the text:

In page 8: “Our data suggests that c-Myc plays a pivotal role in modulating global transcriptional dynamics, subsequently influencing cyclin D expression and CDK4/6 activity.”

e) In neutrophils, neurons and adipocytes, Myc is downregulated and CIP/KIP are upregulated. Already known. Suppressing Myc and overexpression of the inhibitors can suppress proliferation despite the loss of RB1.

Response to e: We appreciate the feedback. Rb protein is a fundamental mediator of cell differentiation and lineage specification¹. Figure 7 illustrates the interplay between the loss of Rb protein and the levels of c-Myc and Cip/Kip during the differentiation process to maintain quiescence. To the best of our knowledge, this has not been well documented in the literature.

271-272: “Cip/Kip family proteins regulate CDK4/6-independent cell-cycle entry after E2F and CDK2 activation triggered by Rb-protein reduction and c-Myc-mediated amplification of transcriptional activity”. “by” makes no sense here.. authors are assuming from the beginning that should be the final conclusion of the paper.

Response: We are grateful to the reviewer for highlighting this important aspect. In response, we have revised our text as follows:

In page 11: “Together, our data imply that Cip/Kip family proteins regulate CDK4/6-independent cell-cycle entry after initial cell-cycle arrest, Rb-protein reduction, and upregulation of mitogenic signaling.”

In general, I am afraid that the manuscript confirms the relevance of several players whose function was already known and proposes a “sequential” effect and a “requirement” for RB1 reduction and Myc activation. Whereas the reduction of RB1 is an interesting observation, it has no functional consequences by itself. All the other assays overexpressing Myc or Cip/Kip etc. are confirmatory of previous data and lack physiological (or pathological) justification as presented.

Reviewer #2 (Remarks to the Author):

I thank authors for responding to my questions raised during the initial review of their paper. While the authors have been able to make their story more accessible for general reader and improve the manuscript during revision, I still have some additional comments and questions about the new data added. I think these points need answers before this manuscript meets the standards of Nature Communications.

Supplementary Figure 2b – In the representative density scatterplot for MCF-10A with Exo-Rb the total % extends over 100% (G0/G1 55%+S 33%+G2 16%). This quantification should be corrected in 2b and calculations changed in 2d.

Supplementary Figure 2d – Y-axis should be renamed as it shows now different cell cycle phases instead of S phase.

Response: We are very grateful to the reviewer for the thorough reading of our manuscript and for pointing out these errors. We have fixed these errors as shown below.

Supplementary Figure 2.

b, c Representative density scatterplot showing Hoechst and EdU staining in MCF-10A (**b**) and RPE1 (**c**) cells without and with YFP-Rb expression ($n = 2,000$ cells/condition).

d Percentage of each cell-cycle phase in MCF-10A and RPE1 cells without and with YFP-Rb expression. Data are shown as mean \pm SD ($n = 3$ biological replicates).

Supplementary Figure 2g and 2h – Can authors explain why there is a small but consistent transition around 13hr point consistently in all heatmaps showing single cell traces for H2B-iRFP, YFP-Rb and NLS-mRuby3.

Response: We appreciate the reviewer for highlighting this crucial point. In our initial experiment, we observed a universal rise across all channels, both in control and CDK4/6i-treatment conditions, roughly 13 hr after the CDK4/6i treatment (below Fig. 1 top). Upon a careful examination of the raw data, we observed a spike in both background and signal levels around this timeframe. To verify this observation, we conducted two additional independent experiments. While each experiment consistently showed the reduction in Rb-protein levels by CDK4/6i treatment, the universal increase observed 13 hr post-treatment in the first experiment was inconsistent in the subsequent independent experiments. Specifically, the second experiment showed this spike around 4 hr post-treatment (below Fig. 1 middle), whereas the third did not present this at all (below Fig. 1 bottom). Given these inconsistencies, we concluded that this surge is not a recurring phenomenon and may be attributed to a technical issue. We are currently working with the microscope company to identify any potential problems. To provide a clarity and avoid misinterpretation, we have updated **Supplementary Fig. 2g, h** using data from the third independent experiment.

Figure 1 for the reviewer.

g, h Heatmap of single-cell traces for H2B-iRFP670, YFP-Rb, and NLS-mRuby3 levels in MCF-10A cells treated with DMSO (g) or palbociclib (1 μ M) (h). Each row represents a single-cell trace over time according to the respective color map.

References for Response to Reviewer Comments:

- 1 Wikenheiser-Brokamp, K. A. Rb family proteins differentially regulate distinct cell lineages during epithelial development. *Development* **131**, 4299-4310 (2004). <https://doi.org/10.1242/dev.01232>

REVIEWERS' COMMENTS

Reviewer #1 (Remarks to the Author):

Despite the previous round of comments, the manuscript does not provide sufficient novel data to merit publication in Nature Communication. No significant changes have been made to the new version. Aside from the lack of novelty, some of the conclusions are not backed up by the data (e.g., sequentiality).

Reviewer #2 (Remarks to the Author):

I thank the authors for their explanations, discussion, and additional experiments to address my previous concerns during the review process. I do not have any further comments/questions. I find this manuscript meets the standards and I support its publication in Nature Communications.

Response to Reviewers:

REVIEWER COMMENTS

Reviewer #1 (Remarks to the Author):

Despite the previous round of comments, the manuscript does not provide sufficient novel data to merit publication in Nature Communication. No significant changes have been made to the new version. Aside from the lack of novelty, some of the conclusions are not backed up by the data (e.g., sequentiality).

Response: We appreciate the reviewer's comments and have taken this opportunity to better highlight the novelty of our study. While the canonical pathway of Rb inactivation is well-documented, the alternative, CDK4/6-independent pathway has not been thoroughly investigated. Our research provides insight into this non-canonical pathway of Rb inactivation and its crosstalk with external signals. Regarding the sequential activation of E2F, our temporal analysis reveals a complex interplay involving c-Myc and Cip/Kip proteins that contribute to the CDK4/6-independent pathway of cell cycle progression. Notably, the modulation of Rb protein levels precedes these events. This finding is pivotal as it elucidates a layered regulatory framework governing E2F activity, thereby advancing our understanding of cell cycle control mechanisms. We believe these clarifications will enhance the comprehension of our study's contribution to the field.

Reviewer #2 (Remarks to the Author):

I thank the authors for their explanations, discussion, and additional experiments to address my previous concerns during the review process. I do not have any further comments/questions. I find this manuscript meets the standards and I support its publication in Nature Communications.

Response: We extend our sincere gratitude to the reviewer for their meticulous and insightful critique during the revision process. The recommendations provided have been invaluable, significantly contributing to the refinement and enhancement of our manuscript. We are confident that these contributions have elevated the overall quality and rigor of our work.